# Cluster-wise Graph Transformer with Dual-granularity Kernelized Attention

**Siyuan Huang**[1,2]    **Yunchong Song**[1]    **Jiayue Zhou**[2]    **Zhouhan Lin**[1*]

[1]LUMIA Lab, Shanghai Jiao Tong University

[2]Paris Elite Institute of Technology, Shanghai Jiao Tong University

`siyuan_huang_sjtu@outlook.com`   `ycsong@sjtu.edu.cn`   `lin.zhouhan@gmail.com`

## Abstract

In the realm of graph learning, there is a category of methods that conceptualize graphs as hierarchical structures, utilizing node clustering to capture broader structural information. While generally effective, these methods often rely on a fixed graph coarsening routine, leading to overly homogeneous cluster representations and loss of node-level information. In this paper, we envision the graph as a network of interconnected node sets without compressing each cluster into a single embedding. To enable effective information transfer among these node sets, we propose the Node-to-Cluster Attention (N2C-Attn) mechanism. N2C-Attn incorporates techniques from Multiple Kernel Learning into the kernelized attention framework, effectively capturing information at both node and cluster levels. We then devise an efficient form for N2C-Attn using the cluster-wise message-passing framework, achieving linear time complexity. We further analyze how N2C-Attn combines bi-level feature maps of queries and keys, demonstrating its capability to merge dual-granularity information. The resulting architecture, Cluster-wise Graph Transformer (Cluster-GT), which uses node clusters as tokens and employs our proposed N2C-Attn module, shows superior performance on various graph-level tasks. Code is available at `https://github.com/LUMIA-Group/Cluster-wise-Graph-Transformer`.

## 1 Introduction

Graph learning represents a rapidly evolving field. Techniques like Graph Neural Networks (GNNs) and Graph Transformers (GT) demonstrate impressive performance across a range of tasks [27, 36, 51], such as social networks [39, 37], time series [22, 30], traffic flow [54, 6] and drug discovery [15, 45]. These methods enhance performance by promoting message propagation at the node level and calculating attention between node pairs, thereby concentrating on node-level interactions.

Recent advancements have extended beyond node-level message propagation, adopting approaches that treat the graph as a hierarchical structure [53, 4, 20], capturing information at multiple levels of the graph [55]. For instance, node clustering pooling segments the graph into multiple clusters [17, 50]. Each cluster is then independently pooled, preserving the structural information of the hierarchical graph. Drawing inspiration from Vision Transformers [7], GraphViT [18] treats subgraphs as tokens and computes attention among them, which enables the model to capture long-distance dependencies and reduces the overall computational complexity compared to node-level Graph Transformers.

However, existing methods based on node clustering rely on a fixed graph coarsening routine [32]. This routine involves partitioning the graph into several clusters and subsequently pooling each cluster into a single node to generate a coarsened version of the original graph. While generally

---

*Zhouhan Lin is the corresponding author

38th Conference on Neural Information Processing Systems (NeurIPS 2024).

effective, research has shown that compressing each cluster into a single embedding can lead to overly uniform cluster representations, which may not accurately reflect the diversity within each cluster [34]. Furthermore, these methods typically simplify the interactions between clusters to basic vertex-level interactions on the coarsened graph. This oversimplification overlooks the rich node-level information contained within each cluster, thereby limiting the potential for richer cluster-wise interactions.

In this work, we propose a different strategy for enhancing cluster-wise interaction. Instead of reducing each cluster to a single node through coarsening, we envision the graph as a network of interconnected node sets. To enable message propagation among these node sets, we develop a method termed Node-to-Cluster Attention (N2C-Attn). N2C-Attn incorporates techniques from Multiple Kernel Learning (MKL) [16] into the kernelized attention framework [46]. By combining kernels at two different granularity levels, N2C-Attn effectively captures hierarchical graph structural information at the cluster level while also preserving node-level details within each cluster.

We propose treating the graph as interconnected node clusters without coarsening, which inherently increases computational complexity. To mitigate this issue, we employ the technique of kernelized softmax [24] to reduce the computational complexity to linear. Consequently, the computation process of N2C-Attn can be viewed as a cluster-wise message propagation: each cluster gathers internal keys and values, then propagates them along weighted edges to the queries of other clusters.

We present a further analysis of how N2C-Attn synthesizes new queries and keys by merging inputs from both node and cluster levels. We consider two scenarios: 1) using the product of kernels and 2) using the convex sum of kernels. The former implicitly conducts a tensor product of the feature maps from both the node-level and cluster-level queries (and keys), adopting this product as the new query (or key) for N2C-Attn. The latter concatenates node and cluster-level feature maps with learnable weights, maintaining their independence and allowing the model to adjust their relative significance. We also demonstrate that cluster-level attention can be regarded as a special case of N2C-Attn.

Our resulting architecture, Cluster-wise Graph Transformer (Cluster-GT), leverages our proposed N2C-Attn module in conjunction with a simple graph partitioning algorithm, Metis [23]. We conduct extensive evaluations of Cluster-GT across eight graph-level datasets, varying in size and domain. Cluster-GT outperforms existing Graph Transformers and graph pooling methods that employ more intricate graph partitioning algorithms, which highlights the effectiveness of enhancing inter-cluster interactions and preserving information at both granular levels. We further analyze the relative weights of the combined kernel, finding that Cluster-GT pays more attention to cluster-level information when handling graphs in the social network domain compared to graphs in the biological domain.

## 2 Background

Consider a graph $\mathcal{G}$ represented by the multi-tuple $(\mathcal{N}, \mathcal{E}, \mathbf{X}, \mathbf{A})$. $\mathcal{N}$ denotes the set of $n$ nodes, $\mathcal{E}$ denotes the set of $m$ edges. $\mathbf{X} \in \mathbb{R}^{n \times d}$ is the feature matrix and $\mathbf{A} \in \mathbb{R}^{n \times n}$ is the adjacency matrix. We use the superscript $P$ to indicate the cluster-level (coarsened) graph: $(\mathcal{N}^{\mathcal{P}}, \mathcal{E}^{\mathcal{P}}, \mathbf{X}^{\mathcal{P}}, \mathbf{A}^{P})$, where $\mathcal{N}^{P}$ represents clusters of nodes, and $\mathcal{E}^{P}$ denotes the edges connecting these clusters.

**Node Clustering Pooling and Cluster Assignment Matrix**   Node clustering pooling captures hierarchical structural information by partitioning and iteratively coarsening the graph to a smaller size [32, 1, 33]. This process involves two main steps. Initially, a Cluster Assignment Matrix (CAM) $\boldsymbol{C} \in \mathbb{R}^{n \times m}$ is generated using a carefully designed strategy, where $n$ represents the number of original nodes, and $m$ indicates the number of clusters. Once the Cluster Assignment Matrix is obtained, it is used to perform graph coarsening, i.e., pooling each cluster into a single node:

$$\boldsymbol{X}^{P} = \boldsymbol{C}^{T}\boldsymbol{X}; \quad \boldsymbol{A}^{P} = \boldsymbol{C}^{T}\boldsymbol{A}\boldsymbol{C} \tag{1}$$

where $\boldsymbol{X}^{P} \in \mathbb{R}^{m \times d}$ and $\boldsymbol{A}^{P} \in \mathbb{R}^{m \times m}$ are the new node features and adjacency matrix, defining the post-coarsening graph structure. $\mathbf{C}_{sj}$ thus represents the weight of the $s$-th node in the $j$-th cluster.

Beyond node clustering pooling, methods exist that leverage node clusters to enhance graph attention [18, 3]. GraphViT [18] utilizes Metis [23] to partition the graph into multiple subgraphs. It then applies mean pooling to each subgraph, treating the pooled clusters as tokens for further attention computation. Despite promising results, GraphViT still adheres to the graph coarsening pipeline, which leads to overly similar cluster representations [34] and the loss of node-level information.

**Generalized Self-attention and Kernelized Softmax** Numerous studies suggest reevaluating the attention mechanism through the lens of kernel methods [46, 24]. The generalized formulation of self-attention utilizes a non-negative kernel function $\kappa(\cdot, \cdot) : \mathbb{R}^{d_k} \times \mathbb{R}^{d_k} \to \mathbb{R}_+$, which can be represented with a corresponding feature map $\phi$. The self-attention mechanism can be expressed as:

$$\text{Attn}(X)_i = \sum_{j=1}^{N} \frac{\kappa(q_i, k_j)}{\sum_{j'=1}^{N} \kappa(q_i, k_{j'})} v_j \tag{2}$$

where $k_j$, $q_i$, and $v_j$ are the corresponding keys, queries, and values. By expressing $\kappa$ with feature map $\kappa(q_i, k_j) = \phi(q_i)^T \phi(k_j)$, the computation simplifies to:

$$\text{Attn}(X)_i = \frac{\phi(q_i) \sum_{j=1}^{N} \phi(k_j)^T v_j}{\phi(q_i) \sum_{j=1}^{N} \phi(k_j)^T} \tag{3}$$

where the sums $\sum_{j=1}^{N} \phi(k_j)^T v_j$ and $\sum_{j=1}^{N} \phi(k_j)^T$ are shared across all nodes and need to be computed only once, thus reducing computational complexity to $\mathcal{O}(N)$ [21]. Various choices of feature maps are shown effective, such as the RBF kernel [46] and Positive Random Features [5].

**Multiple Kernel Learning** The selection of an optimal kernel function $\kappa(\cdot, \cdot)$ is critical for enhancing the performance of kernel-based learning methods. Multiple Kernel Learning (MKL) methods [56, 44] leverage a combination of kernel functions to integrate various features from different perspectives. The resultant kernel, $\kappa_\eta$, is mathematically defined as:

$$\kappa_\eta(\{\mathbf{x}^m\}_{m=1}^M, \{\mathbf{y}^m\}_{m=1}^M) = f_\eta(\{\kappa_m(\mathbf{x}^m, \mathbf{y}^m)\}_{m=1}^M) \tag{4}$$

where $f_\eta : \mathbb{R}^M \to \mathbb{R}$ can be either a linear or nonlinear function. Each $\kappa_m : \mathbb{R}^{D_m} \times \mathbb{R}^{D_m} \to \mathbb{R}$ is a valid kernel for vectors $\mathbf{x}^m, \mathbf{y}^m \in \mathbb{R}^{D_m}$, with $D_m$ representing the dimensionality of each feature. There are various strategies for combining kernels, which represent a dynamic area of research. [16]

In this work, we concentrate on the pairwise scenario, where $M = 2$. We note the two input spaces as $\mathcal{X}$ and $\mathcal{X}'$. For constructing pairwise kernels when elements of each pair belong to different input spaces, we select two fundamental strategies: the tensor product of kernels and the convex linear combination of kernels, which are commonly used on the product space $\mathcal{X} \times \mathcal{X}'$.

Given two kernels $\kappa_1 : \mathcal{X} \times \mathcal{X} \to \mathbb{R}$ and $\kappa_2 : \mathcal{X}' \times \mathcal{X}' \to \mathbb{R}$, the tensor product method is defined as:

$$\kappa_\eta((x, x'), (y, y')) = \kappa_1(x, y) \cdot \kappa_2(x', y') \tag{5}$$

where $(x, x')$, $(y, y')$ are pairs of objects from $\mathcal{X} \times \mathcal{X}'$. While the convex linear combination method is defined as:

$$\kappa_\eta((x, x'), (y, y')) = \alpha \kappa_1(x, y) + \beta \kappa_2(x', y') \tag{6}$$

where $\alpha, \beta \geq 0$ and $\alpha + \beta = 1$. $\alpha$ and $\beta$ are coefficients that balance the contribution of each kernel.

## 3 Node-to-Cluster Attention

In this section, we present the Node-to-Cluster Attention (N2C-Attn) mechanism. We begin in Section 3.1 by defining the concept of N2C-Attn. We then proceed to Section 3.2, where we devise an efficient form of N2C-Attn with the message-passing framework. In Section 3.3, we re-examine N2C-Attn, focusing on the integration of feature maps of queries and keys across two granularities.

### 3.1 Definition of Node-to-Cluster Attention

Node-to-Cluster Attention marks a departure from the graph coarsening pipeline that typically coarsens each cluster into a single embedding. Instead, as shown in Figure 1, we maintain the clusters uncompressed and use N2C-Attn to propagate messages among the inter-connected node clusters. The definition of N2C-Attn is based on the Cluster Assignment Matrix $\mathbf{C}$, which can be obtained through various graph partitioning methods [32]. N2C-Attn focuses on the "post-clustering" phase.

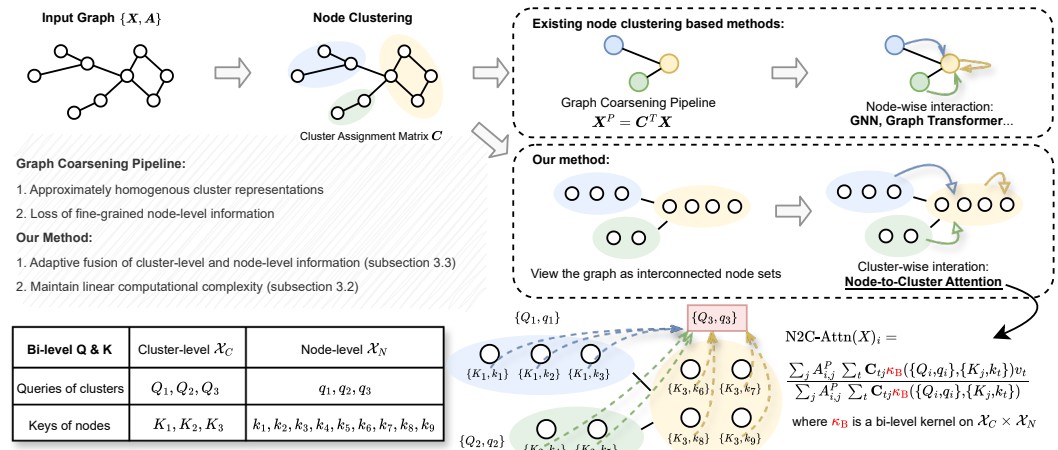

Figure 1: Definition of Node-to-Cluster Attention (N2C-Attn). N2C-Attn perceives the graph as interconnected node sets instead of coarsening each cluster into a single node. It integrates multiple kernel learning methods into the kernelized attention framework to facilitate message propagation among node clusters, simultaneously capturing both the node-level and cluster-level information.

**Bi-level Query and Key**    A key observation is that after node clustering, each node possesses two tiers of information: 1) its individual node feature and 2) the collective feature of its cluster. An effective attention mechanism needs to accommodate these two distinct levels of information. Thus, the $t$-th node in the $j$-th cluster is characterized by a bi-level pair of keys: $\{K_j, k_t\} \in \mathcal{X}_C \times \mathcal{X}_N$:

$$k_t = \mathbf{W}_k h_t, \ K_j = \mathbf{W}'_k \left( \sum_s \mathbf{C}_{sj} h_s \right) \tag{7}$$

where $h_t$ is the feature of the $t$-th node. $k_t \in \mathcal{X}_N$ is the node-level key, which is solely derived from the embedding of $t$-th node, and $K_j \in \mathcal{X}_C$ represents the cluster-level key, which depends on all nodes within the $j$-th cluster. $\mathbf{W}_k$ and $\mathbf{W}'_k$ are two different projections to $\mathcal{X}_N$ and $\mathcal{X}_C$, respectively.

Since we are considering the attention between clusters and nodes, each cluster needs to provide a corresponding bi-level query. Thus, the $i$-th cluster is characterized by a bi-level pair of queries:

$$q_i = \mathbf{W}_v \left( \sum_s \mathbf{C}_{si} h_s \right), \ Q_i = \mathbf{W}'_v \left( \sum_s \mathbf{C}_{si} h_s \right) \tag{8}$$

where $Q_i$ denotes the cluster-level query, $q_i$ denotes the node-level query. $\mathbf{W}_v$ and $\mathbf{W}'_v$ are two different projections to $\mathcal{X}_N$ and $\mathcal{X}_C$, respectively. The bi-level query is thus $\{Q_i, q_i\} \in \mathcal{X}_C \times \mathcal{X}_N$.

Note that we use uppercase letters to represent cluster-level queries and keys, e.g., $\{Q_i, K_j\}$, and lowercase letters to represent node-level queries and keys, e.g., $\{q_i, k_t\}$.

**General Definition of Node-to-Cluster Attention**    Having obtained the bi-level queries and keys, we consider how to use kernels to measure their similarity. We denote a valid kernel in the cluster-level space $\mathcal{X}_C$ as $\kappa_C$, and a valid kernel in the node-level space $\mathcal{X}_N$ as $\kappa_N$. We now consider how to construct a kernel $\kappa_B$ on the tensor product space $\mathcal{X}_C \times \mathcal{X}_N$. $\kappa_B$ stands for **B**i-level kernel.

Given $\{Q_i, q_i\}$, the bi-level query for the $i$-th node cluster, and $\{K_j, k_t\}$, the bi-level key for the $t$-th node in the $j$-th node cluster, the general Node-to-Cluster Attention for the $i$-th cluster is defined as:

$$\text{N2C-Attn}(X)_i = \frac{\sum_j \mathbf{A}^P_{i,j} \sum_t \mathbf{C}_{tj} \kappa_B(\{Q_i, q_i\}, \{K_j, k_t\})}{\sum_j \mathbf{A}^P_{i,j} \sum_t \mathbf{C}_{tj} \kappa_B(\{Q_i, q_i\}, \{K_j, k_t\})} \tag{9}$$

Equation 9 depicts the process of the $i$-th cluster gathering information from nodes of all connected clusters. The attention score between the $i$-th cluster and the $t$-th node in the $j$-th cluster is $\frac{\mathbf{A}^P_{i,j} \mathbf{C}_{tj} \kappa_B(\{Q_i, q_i\}, \{K_j, k_t\})}{\sum_j \mathbf{A}^P_{i,j} \sum_t \mathbf{C}_{tj} \kappa_B(\{Q_i, q_i\}, \{K_j, k_t\})}$. $\kappa_B$ plays a pivotal role in integrating information across cluster and node levels. As described in Section 2, we mainly consider two options for $\kappa_B$: the tensor product method and the linear combination method. Next, we introduce these two different implementations.

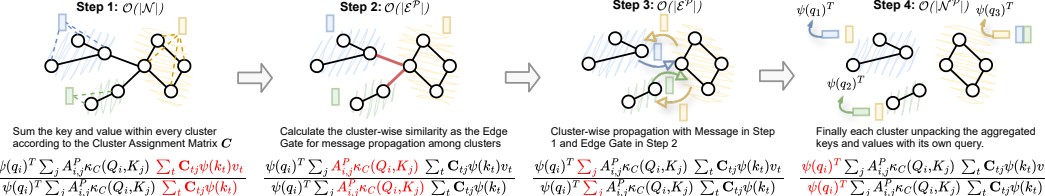

Figure 2: An efficient implementation of N2C-Attn-T with the message-passing framework. $|\mathcal{N}^{\mathcal{P}}|$ denotes the number of clusters and $|\mathcal{E}^{\mathcal{P}}|$ denotes the number of edges between clusters. The computation can be decomposed into 4 steps: 1) aggregation of node-level keys and values within each cluster, 2) computation of gate on each edge with the cluster-level kernel, 3) message propagation among clusters, 4) dot product of aggregated value with the node-level query of each cluster.

**Node-to-Cluster Attention with Tensor Product of Kernels (N2C-Attn-T)**  With the help of Equation 5, we can define the bi-level kernel $\kappa_B$ as:

$$\kappa_{\mathrm{B}}(\{Q_i, q_i\}, \{K_j, k_t\}) = \kappa_C(Q_i, K_j)\kappa_N(q_i, k_t) \tag{10}$$

We can thus rewrite the Node-to-Cluster Attention defined in Equation 9 as:

$$\text{N2C-Attn-T}(X)_i = \frac{\sum_j \mathbf{A}_{i,j}^P \sum_t \mathbf{C}_{tj}\kappa_C(Q_i, K_j)\kappa_N(q_i, k_t)v_t}{\sum_j \mathbf{A}_{i,j}^P \sum_t \mathbf{C}_{tj}\kappa_C(Q_i, K_j)\kappa_N(q_i, k_t)} \tag{11}$$

N2C-Attn-**T** stands for Node-to-Cluster Attention with **T**ensor Product of Kernels. By performing the product between $\kappa_C$ and $\kappa_N$, this construction enables interaction across all dimensions of the feature vectors at different granular levels, thereby capturing the dependencies within the combined feature space. We offer a more detailed explanation in subsection 3.3.

**Node-to-Cluster Attention with Convex Linear Combination of Kernels (N2C-Attn-L)**  With the help of Equation 6, we can also define the bi-level kernel $\kappa_B$ as:

$$\kappa_{\mathrm{B}}(\{Q_i, q_i\}, \{K_j, k_t\}) = \alpha\kappa_C(Q_i, K_j) + \beta\kappa_N(q_i, k_t) \tag{12}$$

where $\alpha, \beta \geq 0$ are learnable parameters and $\alpha + \beta = 1$. We can thus rewrite Equation 9 as:

$$\text{N2C-Attn-L}(X)_i = \frac{\sum_j \mathbf{A}_{i,j}^P \sum_t \mathbf{C}_{tj}(\alpha\kappa_C(Q_i, K_j) + \beta\kappa_N(q_i, k_t))v_t}{\sum_j \mathbf{A}_{i,j}^P \sum_t \mathbf{C}_{tj}(\alpha\kappa_C(Q_i, K_j) + \beta\kappa_N(q_i, k_t))} \tag{13}$$

N2C-Attn-**L** stands for Node-to-Cluster Attention with Convex **L**inear Combination of Kernels. By combining the kernels $\kappa_C$ and $\kappa_N$ with coefficients $\alpha$ and $\beta$, this construction allows for flexible integration of the similarities measured in $\mathcal{X}_C$ and $\mathcal{X}_N$, letting the combined kernel adaptively scale the influence of the cluster-level and node-level information on the overall similarity measure.

### 3.2  Efficient Implementation of Node-to-Cluster Attention

N2C-Attn requires the computation of similarity between queries and keys at two different levels of granularity. Normally, this necessitates a computational complexity of $\mathcal{O}(|\mathcal{N}||\mathcal{N}^P|)$, where $|\mathcal{N}|$ denotes the number of nodes and $|\mathcal{N}^P|$ denotes the number of clusters. To speed up this process, we devise a linear algorithm using the feature map and the message-passing framework. In this subsection, we focus on the efficient implementation of N2C-Attn-T. Following a similar method, we can also develop an efficient implementation for N2C-Attn-L, which is detailed in Appendix A.

To accelerate Equation 11, a key observation is that $\kappa_C$ is correlated to the edges between clusters, serving as the gates on the edges. While $\kappa_N$ involves queries of clusters and keys of nodes. Therefore, we propose separating the node-level and cluster-level computation of $\kappa_N$, and then turning the computation of N2C-Attn-T into a cluster-wise message propagation. We represent $\kappa_N$ using the corresponding feature map: $\kappa_N(q_i, k_t) = \psi(q_i)^T\psi(k_t)$. Thus, Equation 11 can be rewritten as:

$$\text{N2C-Attn-T}(X)_i = \frac{\psi(q_i)^T \sum_j \mathbf{A}_{i,j}^P \kappa_C(Q_i, K_j) \sum_t \mathbf{C}_{tj}\psi(k_t)v_t}{\psi(q_i)^T \sum_j \mathbf{A}_{i,j}^P \kappa_C(Q_i, K_j) \sum_t \mathbf{C}_{tj}\psi(k_t)} \tag{14}$$

With Equation 14, we observe that the computation of N2C-Attn-T can be encompassed within the message-passing framework. Figure 2 shows the complete process, which contains four steps: 1) aggregating node-level keys and values within each cluster, 2) calculating the gate on each edge using the cluster-level kernel, 3) propagating messages among clusters, and 4) computing the dot product of the aggregated value with the node-level query for each cluster. N2C-Attn-T can thus be seen as a form of cluster-wise message propagation. Each cluster acts as a sender, propagating the packaged keys and values of its internal nodes; it also acts as a receiver, using its own query to interpret the information from the received keys and values. The overall time complexity is thus reduced to linear.

### 3.3 Equivalent Feature Maps of Bi-level Kernels

In this subsection, we delve into how the Node-to-Cluster Attention mechanism integrates information across cluster and node levels through the lens of feature maps. We note $\Phi_{\mathrm{B}}$ as the feature map of $\kappa_{\mathrm{B}}$: $\kappa_{\mathrm{B}}(\{Q_i, q_i\}, \{K_j, k_t\}) = \langle \Phi_{\mathrm{B}}(\{Q_i, q_i\}), \Phi_{\mathrm{B}}(\{K_j, k_t\}) \rangle$ where $\langle \cdot, \cdot \rangle$ represents the inner product in the corresponding feature space. Equation 9 can thus be expressed as:

$$\text{N2C-Attn}(X)_i = \frac{\sum_j \mathbf{A}_{i,j}^P \sum_t \mathbf{C}_{tj} \langle \Phi_{\mathrm{B}}(\{Q_i, q_i\}), \Phi_{\mathrm{B}}(\{K_j, k_t\}) \rangle v_t}{\sum_j \mathbf{A}_{i,j}^P \sum_t \mathbf{C}_{tj} \langle \Phi_{\mathrm{B}}(\{Q_i, q_i\}), \Phi_{\mathrm{B}}(\{K_j, k_t\}) \rangle} \tag{15}$$

$\Phi_{\mathrm{B}}(Q_i, q_i)$ represents the feature vector of the newly formulated bi-level query, while $\Phi_{\mathrm{B}}(K_j, k_t)$ represents the feature vector of the newly formulated bi-level key. We are interested in their relationship with the original queries and keys $\{Q_i, q_i, K_j, k_t\}$. We establish the following relationships:

**Proposition 1** *If $\kappa_C(Q_i, K_j) = \langle \phi(Q_i), \phi(K_j) \rangle$ and $\kappa_N(q_i, k_t) = \langle \psi(q_i), \psi(k_t) \rangle$, where $\phi$ and $\psi$ are feature maps for the respective kernels, then the Node-to-Cluster Attention with the tensor product kernel implies the following equivalent feature map:*

$$\Phi_{\mathrm{B}}(\{Q_i, q_i\}) = \phi(Q_i) \otimes \psi(q_i); \quad \Phi_{\mathrm{B}}(\{K_j, k_t\}) = \phi(K_j) \otimes \psi(k_t) \tag{16}$$

*where $\otimes$ represents the outer product of the node-level and cluster-level feature maps. Conversely, the Node-to-Cluster Attention with the convex sum implies the following equivalent feature map:*

$$\Phi_{\mathrm{B}}(\{Q_i, q_i\}) = \sqrt{\alpha} \phi(Q_i) \oplus \sqrt{\beta} \psi(q_i); \quad \Phi_{\mathrm{B}}(\{K_j, k_t\}) = \sqrt{\alpha} \phi(K_j) \oplus \sqrt{\beta} \psi(k_t) \tag{17}$$

*where $\oplus$ represents the concatenation of the weighted node-level and cluster-level feature maps.*

This proposition provides an intuitive understanding of N2C-Attn: by integrating queries and keys from both node-level and cluster-level, N2C-Attn synthesizes new queries and keys enriched with bi-level information. Specifically, using the product of kernels, as detailed in Equation 16, N2C-Attn-T implicitly performs a tensor product between the feature maps of the node-level query (key) and the cluster-level query (key), and finally using the product as the new query (key). This resulting equivalent feature map thus extends into a higher-dimensional space, offering a feature fusion of bi-level information. It's worth noting that we do not need to actually compute the tensor product between the cluster-level and node-level queries or keys, which requires high spatial complexity.

While employing the convex sum of kernels, as detailed in Equation 17, can be regarded as a concatenation of the feature maps of the original node-level and cluster-level queries (keys), appending learnable weights. This approach preserves the independence of queries (keys) at different levels, empowering the model to adjust their relative significance. Besides, we can leverage this point to design an efficient implementation method for N2C-Attn-L. We introduce it in detail in Appendix A.

We offer a further analysis by comparing the assigned attention scores between N2C-Attn and previous cluster-level attention methods. We prove that the attention mechanism used in GraphViT [18], which is based on the graph coarsening pipeline and serves as a cluster-level attention mechanism, can be seen as a special case of our proposed N2C-Attn. More details can be found in Appendix B.

## 4 Cluster-wise Graph Transformer

In this section, we introduce a simple yet performant architecture named Cluster-wise Graph Transformer (Cluster-GT) which takes node clusters as tokens and utilizes N2C-Attn defined in Section 3 to propagate information among clusters. Cluster-GT can be divided into three main modules: 1) a node-level convolution module, 2) a graph partition module, and 3) a cluster-wise interaction module.

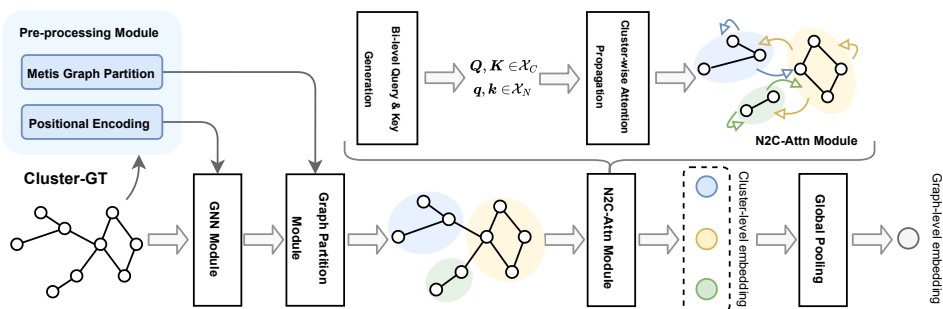

Figure 3: Architecture of Cluster-wise Graph Transformer (Cluster-GT), which can be decomposed into three main modules: 1) a node-wise convolution module with GNN, 2) a graph partition module with Metis, and 3) a cluster-wise interaction module with N2C-Attn.

Figure 3 presents the overall architecture of our proposed Cluster-GT. We begin with a node-level convolution module to capture the local structural information. We try two common options, GCN [27] and GIN [49], during our implementation. We also utilize two graph positional encoding strategies, random-walk structural encoding (RWSE) [9] and Laplacian eigenvector encodings [8], to enhance the perception of the graph structure. More details can be found in Appendix D. For the graph partition module, we use a relatively simple graph partition algorithm, Metis [23], to assign nodes to different clusters. After node clustering assignment, we introduce our proposed N2C-Attn as the cluster-wise interaction module, which propagates information among clusters. This process is divided into two steps: we first calculate the corresponding bi-level keys and queries, and then execute the efficient algorithm of N2C-Attn introduced in subsection 3.2, which outputs a single embedding for each cluster. We finally perform average pooling to obtain the graph-level embedding.

The choice of kernel and feature map is not the main focus of our work. In our implementation, we use the common exp-dot-product $\exp\left(\frac{Q^T K}{\sqrt{d^k}}\right)$ as $\kappa_C$. For the feature map of $\kappa_N$, we try two basic options: $\psi(x) = \mathrm{Elu}(x) + 1$ [40] and $\psi(x) = \mathrm{Relu}(x)$ [24], which we set as a hyperparameter.

Cluster-GT, in conjunction with N2C-Attn, is designed to enhance information exchange between node clusters after the graph partitioning. This process can be viewed as a "post-partitioning" phase, which is a key distinction from many other node-clustering-based methods that primarily focus on optimizing the graph partition itself. In our implementation, we utilize a non-learnable and rigid graph partitioning algorithm, Metis. Notably, the Graph Partition module in Cluster-GT can be replaced with other learnable or flexible graph partitioning strategies, allowing for potential enhancements.

## 5 Evaluation

To evaluate the performance of Cluster-GT, we compare it against two categories of methods: Graph Pooling and Graph Transformers. We conduct experiments on eight graph classification datasets from different domains, including social networks and biology. We further visualize the weight coefficients of the cluster-level and node-level kernels in N2C-Attn-L to observe how the model focuses on different information granularities across different datasets. Additionally, we perform an ablation study, restricting the attention mechanism to different granularities, to demonstrate the benefits of integrating both levels of information. We finally carry out an efficiency study of Cluster-GT. All experiments are conducted on NVIDIA RTX 3090s with 24GB of RAM. Detailed dataset information is available in Appendix E, and more details of the implementation are provided in Appendix F.

### 5.1 Comparison with Graph Pooling Methods

Given the close relationship between Cluster-GT and node clustering methods, we compare Cluster-GT with mainstream Graph Pooling methods:two well-known GNN baselines: GCN [27], GIN [49], six hierarchical pooling approaches: DiffPool [53], SAGPool(H) [29], TopKPool [13], ASAP [42], MinCutPool [4], SEP [50] and five global pooling techniques: Set2Set [48], SortPool [57], SAG-Pool(G) [29], StructPool [55], GMT [2] We test Cluster-GT on six TU datasets [38]: IMDB-BINARY, IMDB-MULTI, COLLAB, MUTAG, PROTEINS, and D&D. The first three datasets are in the field

Table 1: Comparison with Graph Pooling Methods on six TU datasets. The shown accuracies (%) are mean and standard deviation over 10 different runs. We highlight the best results.

| Model | IMDB-BINARY | IMDB-MULTI | COLLAB | MUTAG | PROTEINS | D&D |
|---|---|---|---|---|---|---|
| GCN | $73.26_{\pm 0.46}$ | $50.39_{\pm 0.41}$ | $80.59_{\pm 0.27}$ | $69.50_{\pm 1.78}$ | $73.24_{\pm 0.73}$ | $72.05_{\pm 0.55}$ |
| GIN | $72.78_{\pm 0.86}$ | $48.13_{\pm 1.36}$ | $78.19_{\pm 0.63}$ | $81.39_{\pm 1.53}$ | $71.46_{\pm 1.66}$ | $70.79_{\pm 1.17}$ |
| Set2Set | $72.90_{\pm 0.75}$ | $50.19_{\pm 0.39}$ | $79.55_{\pm 0.39}$ | $69.89_{\pm 1.94}$ | $73.27_{\pm 0.85}$ | $71.94_{\pm 0.56}$ |
| SortPool | $72.12_{\pm 1.12}$ | $48.18_{\pm 0.83}$ | $77.87_{\pm 0.47}$ | $71.94_{\pm 3.55}$ | $73.17_{\pm 0.88}$ | $75.58_{\pm 0.72}$ |
| SAGPool(G) | $72.16_{\pm 0.88}$ | $49.47_{\pm 0.56}$ | $78.85_{\pm 0.56}$ | $76.78_{\pm 2.12}$ | $72.02_{\pm 1.01}$ | $71.54_{\pm 0.91}$ |
| StructPool | $72.06_{\pm 0.64}$ | $50.23_{\pm 0.53}$ | $77.27_{\pm 0.51}$ | $79.50_{\pm 0.75}$ | $75.16_{\pm 0.86}$ | $78.45_{\pm 0.40}$ |
| GMT | $73.48_{\pm 0.76}$ | $50.66_{\pm 0.82}$ | $80.74_{\pm 0.54}$ | $83.44_{\pm 1.33}$ | $75.09_{\pm 0.59}$ | $78.72_{\pm 0.59}$ |
| DiffPool | $73.14_{\pm 0.70}$ | $51.31_{\pm 0.72}$ | $78.68_{\pm 0.43}$ | $79.22_{\pm 1.02}$ | $73.03_{\pm 1.00}$ | $77.56_{\pm 0.64}$ |
| SAGPool(H) | $72.55_{\pm 1.28}$ | $50.23_{\pm 0.44}$ | $78.03_{\pm 0.31}$ | $73.67_{\pm 4.28}$ | $71.56_{\pm 1.49}$ | $74.72_{\pm 0.82}$ |
| TopKPool | $71.58_{\pm 0.95}$ | $48.59_{\pm 0.72}$ | $77.58_{\pm 0.85}$ | $67.61_{\pm 3.36}$ | $70.48_{\pm 1.01}$ | $73.63_{\pm 0.55}$ |
| ASAP | $72.81_{\pm 0.50}$ | $50.78_{\pm 0.75}$ | $78.64_{\pm 0.50}$ | $77.83_{\pm 1.49}$ | $73.92_{\pm 0.63}$ | $76.58_{\pm 1.04}$ |
| MinCutPool | $72.65_{\pm 0.75}$ | $51.04_{\pm 0.70}$ | $80.87_{\pm 0.34}$ | $79.17_{\pm 1.64}$ | $74.72_{\pm 0.48}$ | $78.22_{\pm 0.54}$ |
| SEP-G | $74.12_{\pm 0.56}$ | $51.53_{\pm 0.65}$ | $\mathbf{81.28_{\pm 0.15}}$ | $85.56_{\pm 1.09}$ | $\mathbf{76.42_{\pm 0.39}}$ | $77.98_{\pm 0.57}$ |
| Cluster-GT | $\mathbf{75.10_{\pm 0.84}}$ | $\mathbf{52.13_{\pm 0.78}}$ | $80.43_{\pm 0.52}$ | $\mathbf{87.11_{\pm 1.37}}$ | $\mathbf{76.48_{\pm 0.86}}$ | $\mathbf{79.15_{\pm 0.63}}$ |

Table 2: Comparison with Graph Transformers on ZINC and MolHIV over 4 different runs of 4 different seeds. We highlight the best results. Missing values from literature are indicated as '-'.

| Model | ZINC (MAE ↓) | MolHIV (ROCAUC ↑) |
|---|---|---|
| GT | $0.226_{\pm 0.014}$ | — |
| GraphiT | $0.202_{\pm 0.011}$ | — |
| Graphormer | $0.122_{\pm 0.006}$ | — |
| GPS | $\mathbf{0.070_{\pm 0.004}}$ | $0.7880_{\pm 0.0101}$ |
| SAN+LapPE | $0.139_{\pm 0.006}$ | $0.7775_{\pm 0.0061}$ |
| Graph MLP-Mixer | $\mathbf{0.073_{\pm 0.001}}$ | $0.7997_{\pm 0.0102}$ |
| Graph ViT | $0.085_{\pm 0.005}$ | $0.7792_{\pm 0.0149}$ |
| Cluster-GT | $\mathbf{0.071_{\pm 0.004}}$ | $\mathbf{0.8093_{\pm 0.0136}}$ |

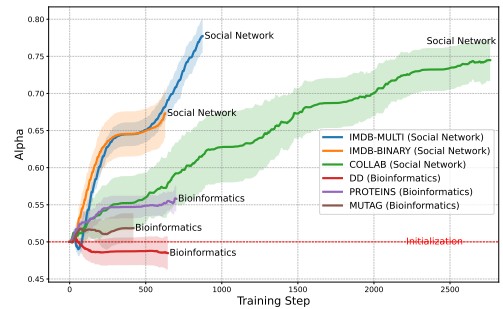

Figure 4: Visualization of $\alpha$ (weight of the cluster-level kernel) during the training process. N2C-Attn learns to pay more attention to cluster-level information in social networks than in bioinformatics.

of social networks, while the latter three are in the field of biology. For a fair comparison, we strictly follow the experimental setup of [50]. Table 1 shows the results, indicating that Cluster-GT outperforms all baselines on most datasets, even though it employs a relatively simple graph partitioning algorithm compared to other node clustering pooling methods. This result highlights the effectiveness of the N2C-Attn module and shows the importance of the interaction between clusters in the "post-partitioning" phase, which is often oversimplified by other node clustering pooling methods.

## 5.2 Comparison with Graph Transformers

To assess the effectiveness of Cluster-GT within the context of Graph Transformers, we compare Cluster-GT with a range of existing Graph Transformers, including GT [8], GraphiT [35], Graphormer [52], GPS [41], SAN+LapPE [28], SAN+RWSE [28], Graph MLP-Mixer [18] and Graph ViT [18]. We conduct the experiment on two datasets: ZINC from Benchmarking GNNs [8] and Mol-HIV from OGB [19]. For a fair comparison, we strictly follow the experimental setup of [18]. The result shown in Figure 2 demonstrates that Cluster-GT surpasses most existing Graph Transformers, underscoring the importance of integrating information at both the cluster and node levels and showcasing the potential of using node clusters as tokens in attention mechanisms.

## 5.3 Visualization of $\alpha$ in N2C-Attn-L

In this subsection, we present the dynamic changes in the weight coefficient $\alpha$ during the training process of N2C-Attn-L. $\alpha$ quantifies the contribution of cluster-level information in the combined kernel, whereas $\beta = 1 - \alpha$ quantifies the node-level information. By enabling the model to

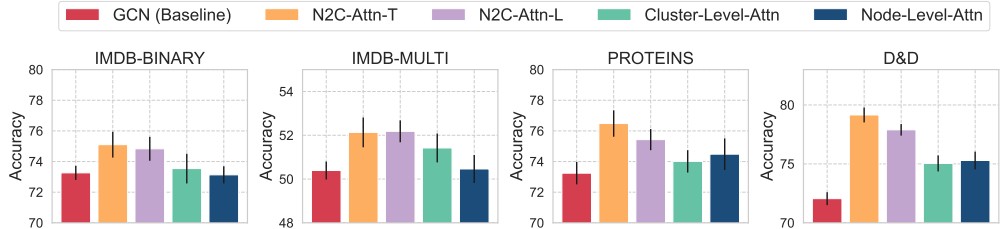

Figure 5: Comparison of different attention strategies. We restrict the attention module in Cluster-GT to focus on different granularities. N2C-Attn-T and N2C-Attn-L represent schemes that integrate information at both the node and cluster granularities. Cluster-Level-Attn focuses solely on cluster-level information, i.e., $\alpha = 1$, while Node-Level-Attn focuses solely on node-level information, i.e., $\alpha = 0$. We provide a detailed description of the methods compared here in subsection F.3.

autonomously learn these coefficients, it dynamically adjusts to the varying importance of information at different granularities. We plot the evolution of $\alpha$ across training steps for six diverse datasets, as shown in Figure 4. We observe that the model automatically adjusts the weights assigned to the two levels of granularity. Notably, for social network datasets, Cluster-GT shows a preference for cluster-level information, whereas, for biology datasets, Cluster-GT balances its attention more equally between both granularities. This result indicates that N2C-Attn has a stronger inclination towards cluster-level information in the social networks domain compared to the biology domain.

### 5.4 Necessity of Combining Cluster-level and Node-level Information

In this subsection, we explore the necessity of fusing kernels of dual granularities within the N2C-Attn module. We analyze four variants: the first two are N2C-Attn-T and N2C-Attn-L, which are the attention schemes utilized in Cluster-GT. N2C-Attn-T deeply integrates cluster-level and node-level information, whereas N2C-Attn-L autonomously adjusts the balance between these two granularities. Then, we create two additional variants that specifically focus on the node level or the cluster level by setting $\alpha$ in N2C-Attn-L to 0 (exclusively focusing on the node-level kernel) and 1 (exclusively focusing on the cluster-level kernel). We provide a detailed description of the methods compared here in subsection F.3. Figure 5 shows the experimental results. We find that the variants that combine attention from both levels significantly surpass those that do not, with N2C-Attn-T leading marginally. This highlights the effectiveness of N2C-Attn's multiple kernel learning approach in integrating diverse levels of information. We reference the performance of GCN from Table 1 as a baseline.

## 6 Other Methods Involving Graph Coarsening

In this section, we will briefly introduce some existing research on GNNs with graph coarsening to capture broader structural information, aside from the node clustering pooling introduced in section 2.

[12] utilizes a dual-graph structure, employing a hierarchical message passing strategy between a molecular graph and its junction tree to facilitate a bidirectional flow of information. This concept of interaction between the coarsened graph (clusters) and the original graph (nodes) is similar to our N2C-Attn. However, the difference lies in [12]'s approach to propagating messages between clusters and nodes, whereas N2C-Attn integrates cluster and node information directly in the attention calculation using a multiple-kernel method. [58] introduces a novel node sampling strategy as an adversarial bandit problem and implements a hierarchical attention mechanism with graph coarsening to efficiently address long-range dependencies. [31] uses graph pooling to coarsen nodes into fewer representatives, focusing attention on these pooled nodes to manage scalability and computational efficiency. [25] introduces the Subgraph-To-Node (S2N) translation method, coarsening subgraphs into nodes to improve subgraph representation learning. [14] introduces HIGH-PPI, a double-viewed hierarchical graph learning model that uses a hierarchical graph combining protein-protein interaction networks and chemically described protein graphs to accurately predict PPIs and interpret their molecular mechanisms. Despite achieving good results in their respective downstream tasks, these methods still follow the graph coarsening pipeline, whereas our work attempts to break this limitation and has demonstrated effectiveness on various graph-level tasks.

# 7 Conclusion

Our Node-to-Cluster Attention mechanism leverages the strengths of both node-level and cluster-level information processing without succumbing to the limitations of the graph coarsening pipeline. By conceptualizing the graph as interconnected node sets and integrating kernelized attention with multiple kernel learning, we effectively bridge the gap between cluster-level and node-level spaces, capturing the hierarchical structure of graphs as well as the node-level information. We develop an efficient form of N2C-Attn using the message-passing framework and techniques of kernelized softmax. Our Cluster-wise Graph Transformer, empowered by a straightforward partitioning strategy and the N2C-Attn module, demonstrates robust performance across diverse graph datasets. Extensive experiments have demonstrated the effectiveness of our Cluster-GT and N2C-Attn modules. We offer a further discussion on the current limitation and potential impact in Appendix H and Appendix I.

## Acknowledgement

This work was sponsored by the National Key Research and Development Program of China (No. 2023ZD0121402) and National Natural Science Foundation of China (NSFC) grant (No.62106143).

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

## A  Efficient implementation of Node-to-Cluster Attention with Convex Linear Combination of Kernels

In this section, we will devise an efficient form for the Node-to-Cluster Attention with Convex Linear Combination of Kernels:

$$\text{N2C-Attn-L}(X)_i = \frac{\sum_j \mathbf{A}^P_{i,j} \sum_t \mathbf{C}_{tj}(\alpha\kappa_C(Q_i, K_j) + \beta\kappa_N(q_i, k_t))v_t}{\sum_j \mathbf{A}^P_{i,j} \sum_t \mathbf{C}_{tj}(\alpha\kappa_C(Q_i, K_j) + \beta\kappa_N(q_i, k_t))} \tag{18}$$

We introduce the corresponding feature map: $\kappa_C(Q_i, K_j) = \langle\phi(Q_i), \phi(K_j)\rangle$ and $\kappa_N(q_i, k_t) = \langle\psi(q_i), \psi(k_t)\rangle$. According to Prop.1 , we have $\Phi_{\mathrm{B}}(\{Q_i, q_i\}) = \sqrt{\alpha}\phi(Q_i) \oplus \sqrt{\beta}\psi(q_i); \quad \Phi_{\mathrm{B}}(\{K_j, k_t\}) = \sqrt{\alpha}\phi(K_j) \oplus \sqrt{\beta}\psi(k_t)$. Thus we can rewrite Equation 18 as:

$$\begin{aligned} \text{N2C-Attn-L}(X)_i &= \frac{\sum_j \mathbf{A}^P_{i,j} \sum_t \mathbf{C}_{tj} \left[\sqrt{\alpha}\phi(Q_i) \oplus \sqrt{\beta}\psi(q_i)\right]^T \left[\sqrt{\alpha}\phi(K_j) \oplus \sqrt{\beta}\psi(k_t)\right] v_t}{\sum_j \mathbf{A}^P_{i,j} \sum_t \mathbf{C}_{tj} \left[\sqrt{\alpha}\phi(Q_i) \oplus \sqrt{\beta}\psi(q_i)\right]^T \left[\sqrt{\alpha}\phi(K_j) \oplus \sqrt{\beta}\psi(k_t)\right]} \\ &= \frac{\left[\sqrt{\alpha}\phi(Q_i) \oplus \sqrt{\beta}\psi(q_i)\right]^T \sum_j \mathbf{A}^P_{i,j} \sum_t \mathbf{C}_{tj} \left[\sqrt{\alpha}\phi(K_j) \oplus \sqrt{\beta}\psi(k_t)\right] v_t}{\left[\sqrt{\alpha}\phi(Q_i) \oplus \sqrt{\beta}\psi(q_i)\right]^T \sum_j \mathbf{A}^P_{i,j} \sum_t \mathbf{C}_{tj} \left[\sqrt{\alpha}\phi(K_j) \oplus \sqrt{\beta}\psi(k_t)\right]} \end{aligned} \tag{19}$$

where $\left[\sqrt{\alpha}\phi(Q_i) \oplus \sqrt{\beta}\psi(q_i)\right]$ is the weighted concatenation of the feature map of bi-level queries, while $\left[\sqrt{\alpha}\phi(K_j) \oplus \sqrt{\beta}\psi(k_t)\right]$ is the weighted concatenation of the feature map of bi-level keys.

Equation 19 allows us to implement N2C-Attn-L with message-passing framework, which is similar to the implementation that we have devised in subsection 3.2. We can first calculate the equivalent feature map of the bi-level query $\left[\sqrt{\alpha}\phi(Q_i) \oplus \sqrt{\beta}\psi(q_i)\right]$ for each cluster, and the equivalent feature map of the bi-level key $\left[\sqrt{\alpha}\phi(K_j) \oplus \sqrt{\beta}\psi(k_t)\right]$ for each node. Then we aggregate the keys of nodes within every cluster respectively to get $\sum_t \mathbf{C}_{tj} \left[\sqrt{\alpha}\phi(K_j) \oplus \sqrt{\beta}\psi(k_t)\right] v_t$ and $\sum_t \mathbf{C}_{tj} \left[\sqrt{\alpha}\phi(K_j) \oplus \sqrt{\beta}\psi(k_t)\right]$. After getting these two aggregated "messages", we perform a message passing according to the adjacency matrix of the coarsened graph $\mathbf{A}^P$. And finally, we unpack the aggregated information by calculating the dot product with the feature map of the bi-level query $\left[\sqrt{\alpha}\phi(Q_i) \oplus \sqrt{\beta}\psi(q_i)\right]^T$. In summary, by using the corresponding feature map and this cluster-level message propagation, we can achieve an implementation method for N2C-Attn-L with linear computational complexity.

## B  Relationship between GraphViT and N2C-Attn mechanism

In this subsection, we present a detailed justification for why the attention mechanism used in GraphViT [18] can be regarded as a special case of our proposed N2C-Attn. Please note that in this section, we mainly focus on the similarities and differences in attention computation between GraphViT and N2C-Attn. While GraphViT also enhances its performance and expressive power through the use of various positional encodings, residual connections, and normalization techniques, these modules are not the primary focus of this section and therefore will not be discussed.

GraphViT first uses Metis to partition the graph (with overlapping nodes), and then performs average pooling within each partition. We denote the embedding of the clusters as $\mathbf{X}^P$, GraphViT then performs the Graph-based Hadamard Attention: $\text{G-Hadamard-Attn}(\mathbf{X}^P)$ to capture the dependencies between the clusters, where G-Hadamard-Attn is defined as $\left(A^P \odot \text{softmax}\left(\frac{QK^T}{\sqrt{d}}\right)\right)V$.

We denote the node set if the $p$-th cluster as $\mathcal{V}_p$, then the average pooling process can be written as: $x_p = \frac{1}{|\mathcal{V}_p|}\sum_{i\in\mathcal{V}_p} x_{i,p}$, where $x_{i,p}$ is the embedding of the $i$-th node within the $p$-th cluster, and $x_p$ is the embedding of the $p$-th cluster. And we denote the connected clusters (cluster-wise neighbors) of

the $i$-th cluster as $\mathcal{N}_i$. Then the Hadamard Attention used in GraphViT can be written as:

$$
\begin{aligned}
\text{G-Hadamard-Attn}(X)_i &= \frac{\sum_{j:\mathcal{V}_j N_i} A^P_{i,j}\, \kappa\,(Q_i, K_j)\, V_j}{\sum_{j:\mathcal{V}_j N_i} A^P_{i,j}\, \kappa\,(Q_i, K_j)} \\
&= \frac{\sum_{j:\mathcal{V}_j \in \mathcal{N}_i} A^P_{i,j}\, \kappa\,(Q_i, K_j)\, \sum_{t \in \mathcal{V}_j} \frac{1}{|\mathcal{V}_j|} v_t}{\sum_{j:\mathcal{V}_j \in \mathcal{N}_i} A^P_{i,j}\, \kappa\,(Q_i, K_j)\, \sum_{t \in \mathcal{V}_j} \frac{1}{|\mathcal{V}_j|}} \\
&= \frac{\sum_{j:\mathcal{V}_j \in \mathcal{N}_i} A^P_{i,j}\, \sum_{t \in \mathcal{V}_j} \kappa\,(Q_i, K_j)\, \frac{1}{|\mathcal{V}_j|} v_t}{\sum_{j:\mathcal{V}_j \in \mathcal{N}_i} A^P_{i,j}\, \sum_{t \in \mathcal{V}_j} \kappa\,(Q_i, K_j)\, \frac{1}{|\mathcal{V}_j|}}
\end{aligned}
\tag{20}
$$

where $\kappa\,(Q_i, K_j) = \exp\left(\frac{Q_i^T K_j}{\sqrt{d}}\right)$. $Q$ and $K$ can be seen as cluster-level queries and keys.

Now, we check the corresponding form of N2C-Attn in this case. Since we use the Metis graph partitioning algorithm, which divides the graph into several separate subgraphs and produces a hard cluster Assignment Matrix:

$$
\mathbf{C}^{\text{Metis}}_{nm} = \begin{cases} \frac{1}{|\mathcal{V}_m|} & \text{if the } n\text{-th node is in the } m\text{-th cluster} \\ 0 & \text{otherwise} \end{cases}
\tag{21}
$$

With the help of Equation 21, we can rewrite Equation 11 as:

$$
\text{N2C-Attn-T}(X)_i = \frac{\sum_{j:\mathcal{V}_j \in \mathcal{N}_i} A^P_{i,j} \sum_{t \in V_j} \kappa_C(Q_i, K_j)\kappa_N(q_i, k_t) v_t}{\sum_{j:\mathcal{V}_j \in \mathcal{N}_i} A^P_{i,j} \sum_{t \in V_j} \kappa_C(Q_i, K_j)\kappa_N(q_i, k_t)}
\tag{22}
$$

Comparing Equation 20 and Equation 22, if we set $\kappa_C(Q_i, K_j) = \kappa\,(Q_i, K_j) = \exp\left(\frac{Q_i^T K_j}{\sqrt{d}}\right)$, then the only difference between these two formulas lies in the coefficient before $v_t$. In fact, Equation 20 can be seen as a special case of Equation 22 where $\kappa_N(q_i, k_t) = \frac{1}{|\mathcal{V}_j|}$.

From our analysis above, the difference between the cluster-level attention used in GraphViT and N2C-Attn is as follows: the former assigns the same weight to all nodes within each cluster $= \frac{1}{|\mathcal{V}_j|}$, while the latter allows different attention weights for the nodes within each cluster and uses a node-level kernel $\kappa_N$ to learn these weights.

For simplicity, we only prove that the cluster-level attention used in GraphViT can be considered a special case of Node-to-Cluster Attention with Tensor Product of Kernels (N2C-Attn-T) in this section. In fact, we can similarly argue that the cluster-level attention used in GraphViT is a special case of Node-to-Cluster Attention with Convex Linear Combination of Kernels (N2C-Attn-L). Their most important difference is that GraphViT still follows the graph coarsening pipeline and only uses cluster-level kernels for attention calculation, whereas N2C-Attn integrates both cluster-level and node-level kernels to perform the attention computation.

## C  Proof of Proposition 1

### C.1  Proof of Equation 16

In this subsection, we offer a detailed proof for Equation 16.

If $Q_i, K_j$ are $d^C$-dimensional vectors from the cluster-level space $\mathcal{X}_C$ and $q_i, k_t$ are $d^N$-dimensional vectors from the node-level space $\mathcal{X}_N$. Consider two kernel functions $\kappa_C, \kappa_N$ from the cluster-level space $\mathcal{X}_C$ and the node-level space $\mathcal{X}_N$ respectively, with the corresponding feature map: $\kappa_C(Q_i, K_j) = \langle \phi(Q_i), \phi(K_j) \rangle$ and $\kappa_N(q_i, k_t) = \langle \psi(q_i), \psi(k_t) \rangle$.

Now we consider the case of Node-to-Cluster Attention with Tensor Product of Kernels, where we use the product of the kernels $\kappa_C, \kappa_N$ to construct the bi-level kernel: $\kappa_{\mathrm{B}}(\{Q_i, q_i\}, \{K_j, k_t\}) = \kappa_C(Q_i, K_j)\kappa_N(q_i, k_t)$ where $\kappa_{\mathrm{B}}$ is the bi-level kernel from the tensor product of two original spaces

$\mathcal{X}_C \times \mathcal{X}_N$. Then, for all $(Q_i, K_j) \in \mathcal{X}_C^2$ and $(q_i, k_t) \in \mathcal{X}_N^2$, we have:

$$\begin{aligned}
\kappa_{\mathrm{B}}(\{Q_i, q_i\}, \{K_j, k_t\}) &= \kappa_C(Q_i, K_j)\kappa_N(q_i, k_t) \\
&= \langle \phi(Q_i), \phi(K_j) \rangle \cdot \langle \psi(q_i), \psi(k_t) \rangle \\
&= \left( \sum_m^{d^C} \phi_m(Q_i)\phi_m(K_j) \right) \cdot \left( \sum_n^{d^N} \psi_n(q_i)\psi_n(k_t) \right) \\
&= \sum_m^{d^C} \sum_n^{d^N} (\phi_m(Q_i)\psi_n(q_i)) \cdot (\phi_m(K_j)\psi_n(k_t))
\end{aligned} \quad (23)$$

Thus, we can construct the following feature map:

$$\Phi_{\mathrm{B}}(u, v) = \begin{bmatrix} \phi_1(u)\psi_1(v) \\ \phi_1(u)\psi_2(v) \\ \phi_2(u)\psi_1(v) \\ \vdots \end{bmatrix} = \phi(u) \otimes \psi(v) \quad (24)$$

where $\Phi_{\mathrm{B}}$ is a $d^C d^N \times 1$ feature map. For each pair $(i, j)$, $\Phi_{\mathrm{B},(i,j)}(u, v) = \phi_i(u)\psi_j(v)$, where $1 \le i \le d^C$ and $1 \le j \le d^N$. This composite feature map $\Phi_{\mathrm{B}}$ corresponds to the kernel $\kappa_{\mathrm{B}}$.

With the composite feature map $\Phi_{\mathrm{B}}$, we can rewrite Equation 23 as:

$$\begin{aligned}
\kappa_{\mathrm{B}}(\{Q_i, q_i\}, \{K_j, k_t\}) &= \sum_m^{d^C} \sum_n^{d^N} \Phi_{\mathrm{B}(m,n)}(Q_i, q_i) \cdot \Phi_{\mathrm{B}(m,n)}(K_j, k_t) \\
&= \langle \Phi_{\mathrm{B}}(Q_i, q_i), \Phi_{\mathrm{B}}(K_j, k_t) \rangle
\end{aligned} \quad (25)$$

which proves Equation 16.

## C.2  Proof of Equation 17

In this subsection, we offer a detailed proof for the Equation 17.

Again, we use $\kappa_C, \kappa_N$ to denote the two kernel functions from the cluster-level space $\mathcal{X}_C$ and the node-level space $\mathcal{X}_N$ respectively, with the corresponding feature map: $\kappa_C(Q_i, K_j) = \langle \phi(Q_i), \phi(K_j) \rangle$ and $\kappa_N(q_i, k_t) = \langle \psi(q_i), \psi(k_t) \rangle$.

Now we consider the case of Node-to-Cluster Attention with Convex Linear Combination of Kernels, where we use the convex linear combination of kernels $\kappa_C, \kappa_N$ to construct the bi-level kernel: $\kappa_{\mathrm{B}}(\{Q_i, q_i\}, \{K_j, k_t\}) = \alpha\kappa_C(Q_i, K_j) + \beta\kappa_N(q_i, k_t)$ where $\alpha, \beta \ge 0$ and $\alpha + \beta = 1$. $\alpha$ and $\beta$ are coefficients that balance the contribution of each kernel. Then, for all $(Q_i, K_j) \in \mathcal{X}_C^2$ and $(q_i, k_t) \in \mathcal{X}_N^2$, we have:

$$\begin{aligned}
\kappa_{\mathrm{B}}(\{Q_i, q_i\}, \{K_j, k_t\}) &= \alpha\kappa_C(Q_i, K_j) + \beta\kappa_N(q_i, k_t) \\
&= \alpha\langle \phi(Q_i), \phi(K_j) \rangle + \beta\langle \psi(q_i), \psi(k_t) \rangle \\
&= \langle \sqrt{\alpha}\phi(Q_i), \sqrt{\alpha}\phi(K_j) \rangle + \langle \sqrt{\beta}\psi(q_i), \sqrt{\beta}\psi(k_t) \rangle \\
&= \left( \sum_m^{d^C} \sqrt{\alpha}\phi_m(Q_i)\sqrt{\alpha}\phi_m(K_j) \right) + \left( \sum_n^{d^N} \sqrt{\beta}\psi_n(q_i)\sqrt{\beta}\psi_n(k_t) \right)
\end{aligned} \quad (26)$$

Thus, we can construct the following feature map:

$$\Phi_{\mathrm{B}}(u, v) = \sqrt{\alpha}\phi(u) \oplus \sqrt{\beta}\psi(v) \quad (27)$$

where $\Phi_{\mathrm{B}}$ is a weighted concatenation of the feature maps $\phi$ and $\psi$. In other words, if the feature maps $\phi$ and $\psi$ have $d^C$ and $d^N$ coordinates respectively, then $\Phi_{\mathrm{B}}$ has $d^C + d^N$ coordinates; for any pair $(u, v) \in \mathcal{X}_C \times \mathcal{X}_N$, the first $d^C$ coordinates of $\Phi_{\mathrm{B}}(u, v)$ are $\sqrt{\alpha}\phi_1(u), \sqrt{\alpha}\phi_2(u), \ldots, \sqrt{\alpha}\phi_{d^C}(u)$ and the remaining $d^N$ coordinates of $\Phi_{\mathrm{B}}(u, v)$ are $\sqrt{\beta}\psi_1(v), \sqrt{\beta}\psi_2(v), \ldots, \sqrt{\beta}\psi_{d^N}(v)$.

With the composite feature map $\Phi_{\mathrm{B}}$, we can rewrite Equation 26 as:

$$\kappa_{\mathrm{B}}(\{Q_i, q_i\}, \{K_j, k_t\}) = \langle \Phi_{\mathrm{B}}(Q_i, q_i), \Phi_{\mathrm{B}}(K_j, k_t) \rangle \quad (28)$$

which proves Equation 17.

Table 3: Summary statistics of datasets

| Dataset | #Graphs | Avg. #Nodes | Avg. #Edges | Task | Metric |
|---------|---------|-------------|-------------|------|--------|
| IMDB-BINARY | 1000 | 19.8 | 96.5 | binary classif. | Accuracy |
| IMDB-MULTI | 1500 | 13.0 | 66.0 | 3-class classif. | Accuracy |
| COLLAB | 5000 | 74.49 | 2457.8 | 3-class classif. | Accuracy |
| MUTAG | 188 | 17.93 | 19.8 | binary classif. | Accuracy |
| PROTEINS | 1113 | 39.1 | 72.8 | binary classif. | Accuracy |
| D&D | 1178 | 284.3 | 715.7 | binary classif. | Accuracy |
| ZINC | 12,000 | 23.2 | 24.9 | regression | MAE |
| MolHIV | 41,127 | 25.5 | 54.9 | binary classif. | ROCAUC |

# D More Details of Cluster-GT

**Positinal Encoding**   Positional encoding in graphs plays a crucial role in providing spatial context to nodes. Following [18], we adopt two different strategies: random-walk structural encoding (RWSE) [9] and Laplacian eigenvector encodings [8]. We concatenate the positional encoding with node features as the model input. Additionally, we have tried the patch-wise positional encoding proposed by [18] and set it as a hyperparameter for the Cluster-GT architecture.

**Node-wise Convolution**   In our Cluster-GT framework, we have experimented with incorporating GCN [27] and GIN [49] as the node-wise convolution modules, setting the choice between them as a hyperparameter to optimize performance. GIN is particularly notable for its ability to improve model expressiveness, which is crucial in distinguishing different graph structures. Moreover, the design of our Cluster-GT framework is modular, allowing the node-wise convolution module to be freely replaced by any other method.

**Bi-level Queries and Keys**   After the node clustering assignment, we obtain various node clusters. When generating cluster-level queries or keys, we have tried two options: 1) using DeepSets, 2) aggregating the queries and keys within a cluster. We set these two options as a hyperparameter for the model architecture. In our experiments, we find that these two options have similar performance. Additionally, we find that having the same cluster-level and node-level queries does not affect the model's performance, as long as the bi-level keys remain different. Therefore, we treat whether the queries at the two levels are identical as an optional hyperparameter.

**Other details**   Just like other Transformer structures [47], we incorporate residual connections between the attention layers and MLP, along with layer normalization to enhance training stability. In N2C-Attn, after outputting representations at each cluster level, we ultimately obtain a graph-level representation through average pooling. For N2C-Attn-T and N2C-Attn-L, we use the efficient algorithms introduced in subsection 3.2 and Appendix A in our implementation, respectively.

# E Dataset Information

Our experiments employ a variety of benchmark datasets commonly used in graph learning research. These datasets are chosen for their distinct characteristics and relevance in testing graph classification algorithms. The summary statistics of datasets are shown in Table 3.

## E.1 Datasets used in subsection 5.1

We organize the datasets into categories based on their application domains. • **Social Networks**: IMDB-BINARY and IMDB-MULTI are derived from the Internet Movie Database (IMDB) and include graphs representing the ego-networks of different movie genres. In IMDB-BINARY, each graph is labeled as either Action or Romance. IMDB-MULTI includes three genres: Comedy, Romance, or Sci-Fi. Nodes represent actors, and edges are placed between nodes if the actors have co-starred in a movie. These datasets are used to evaluate the capability of graph classification models in social network analysis. • **Scientific Collaboration Networks**: COLLAB represents the ego-collaboration networks of researchers from three fields: High Energy Physics, Condensed Matter

Physics, or Astrophysics. Nodes represent scientists, and edges are drawn between scientists who have co-authored papers. COLLAB tests the model's ability to recognize different collaborative patterns in scientific domains. • **Biochemical Molecules**: For the biochemical domain, we have chosen three datasets: MUTAG, PROTEINS and D&D. MUTAG comprises 188 chemical compounds represented as graphs, where nodes symbolize atoms and edges denote chemical bonds. Each graph is labeled based on its mutagenic effect on bacteria, serving as a benchmark for bioinformatics applications in predicting chemical properties. PROTEINS consists of protein structures, where each graph corresponds to a protein, nodes to secondary structure elements (SSEs), and edges connect nodes if they are adjacent either in the amino acid sequence or in 3D space. Proteins are classified into enzymes or non-enzymes, providing a basis for studying complex biological structures. D&D includes protein structures with nodes representing amino acids and edges based on spatial closeness. Graphs are labeled according to whether the protein is associated with a disease, challenging the algorithms to decode intricate biological interactions. These datasets collectively provide a comprehensive suite for evaluating across different real-world scenarios.

### E.2 Datasets used in subsection 5.2

Here, we select two datasets used in biochemical molecule and drug discovery research: • **Biochemical Molecules**: The ZINC dataset is a collection of chemical compounds that are representative of real-world molecular data. This dataset is utilized predominantly for regression tasks such as predicting the scalar measure of molecular solubility. Each compound is represented as a graph where nodes are atoms and edges are chemical bonds, making it crucial for testing the accuracy of models in predicting molecular attributes. • **Drug Discovery**: The MolHIV dataset is part of the MoleculeNet suite, specifically designed for binary classification tasks related to HIV drug activity. Graphs in this dataset represent molecular structures where nodes are atoms and edges correspond to bonds. The task is to predict whether a molecule inhibits the HIV virus, which is vital for speeding up the discovery of potential therapeutic agents.

## F Implementation Details

### F.1 Introduction of baselines

**Baselines for subsection 5.1** Initially, we utilize two well-known GNN architectures for comparison: GCN [27] and GIN [49]. Subsequently, we incorporate six hierarchical pooling approaches as baselines: DiffPool [53], SAGPool(H) [29], TopKPool [13], ASAP [42], MinCutPool [4] and SEP [50]. In addition to these hierarchical pooling methods, considerable attention has been given to global pooling strategies for graph classification. Therefore, we also evaluate five global pooling techniques: Set2Se [48], SortPool [57], SAGPool(G) [29], StructPool [55], and GMT [2] for comparative analysis.

**Baselines for subsection 5.2** Next, we compare Cluster-GT against popular Graph Transformers with SOTA results, including GT [8], GraphiT [35], Graphormer [52], GPS [41], SAN+LapPE [28], SAN+RWSE [28]. These models represent cutting-edge advancements in graph neural network technology, each introducing unique methods to handle graph-structured data effectively.

### F.2 Experimental Details

The model is implemented using PyTorch and PyG [11]. Experiments are conducted on NVIDIA RTX 3090 GPUs. For optimization, the Adam [26] optimizer is utilized, adhering to the default settings of $\beta_1 = 0.9$, $\beta_2 = 0.999$, and $\varepsilon = 1e^-8$.

**Experimental Details of subsection 5.1** The model's performance is assessed using a 10-fold cross-validation approach, with dataset splits adhering to the standard established training/test partitions [50]. Moreover, 10 percent of the training data is allocated as validation data to ensure a fair comparison, as per [10]. The initial feature inputs are aligned with the fair comparison setting [10]. An early stopping criterion is implemented, halting training if there is no improvement in validation loss over 50 epochs. The training process is capped at a maximum of 500 epochs. The average performance on the test sets is reported after conducting the experiments 10 times.

**Experimental Details of subsection 5.2**   Each experiment is run with four different seeds, and the averaged results are reported from the epoch that achieved the best validation metric. We use a batch size of 64. We utilize a standard train/validation/test dataset split following [18].

### F.3   Attention strtegies compared in subsection 5.4

In this section, we provide a detailed introduction to the four different node-to-clustering attention strategies: 'N2C-Attn-T','N2C-Attn-L','Cluster-Level-Attn','Node-Level-Attn', compared in the ablation study. Note that for these four variants, we simply replaced the attention mechanism in Cluster-GT without modifying any other parts of the model.

- *N2C-Attn-T*:

$$\text{N2C-Attn-T}(X)_i = \frac{\sum_j \mathbf{A}_{i,j}^P \sum_t \mathbf{C}_{tj} \kappa_C(Q_i, K_j) \kappa_N(q_i, k_t) v_t}{\sum_j \mathbf{A}_{i,j}^P \sum_t \mathbf{C}_{tj} \kappa_C(Q_i, K_j) \kappa_N(q_i, k_t)} \tag{29}$$

- *N2C-Attn-L*:

$$\text{N2C-Attn-L}(X)_i = \frac{\sum_j \mathbf{A}_{i,j}^P \sum_t \mathbf{C}_{tj} (\alpha \kappa_C(Q_i, K_j) + \beta \kappa_N(q_i, k_t)) v_t}{\sum_j \mathbf{A}_{i,j}^P \sum_t \mathbf{C}_{tj} (\alpha \kappa_C(Q_i, K_j) + \beta \kappa_N(q_i, k_t))} \tag{30}$$

- *Cluster-Level-Attn*:

$$\text{Cluster-Level-Attn}(X)_i = \frac{\sum_j \mathbf{A}_{i,j}^P \sum_t \mathbf{C}_{tj} \kappa_C(Q_i, K_j) v_t}{\sum_j \mathbf{A}_{i,j}^P \sum_t \mathbf{C}_{tj} \kappa_C(Q_i, K_j)} \tag{31}$$

- *Node-Level-Attn*:

$$\text{Node-Level-Attn}(X)_i = \frac{\sum_j \mathbf{A}_{i,j}^P \sum_t \mathbf{C}_{tj} \kappa_N(q_i, k_t) v_t}{\sum_j \mathbf{A}_{i,j}^P \sum_t \mathbf{C}_{tj} \kappa_N(q_i, k_t)} \tag{32}$$

It is worth noting that *Cluster-Level-Attn* is a special case of *N2C-Attn-L* when the cluster-level coefficient $\alpha = 1$, while *Node-Level-Attn* is also a special case of *N2C-Attn-L* when the node-level coefficient $\beta = 1 - \alpha = 1$.

## G   On the Naming Issue Between RWSE and RWPE

We use the term "Random Walk Positional Encoding (RWPE)" in this work. In this section, we will briefly discuss the naming issue between RWSE and RWPE. In the original paper that introduced RWPE [9], the term "Random Walk Positional Encoding (RWPE)" was proposed. This paper utilized the self-landing probability of nodes in a random walk to capture neighborhood structural information.

Subsequently, an influential work in the graph transformer domain [41] made a clear distinction between two types of encodings for structure and position, naming them Positional Encoding (PE) and Structural Encoding (SE). Positional encodings are intended to provide an understanding of a node's position within the graph, while structural encodings aim to embed the structure of graphs or subgraphs, enhancing the expressivity and generalizability of GNNs.

Interestingly, [41] argues that the Random Walk Positional Encoding (RWPE) proposed in [9] actually serves as a Structural Encoding (SE). Based on our investigation, it is likely that [41] began using the term RWSE instead of RWPE. Many subsequent studies, likely influenced by [41], such as [43, 18], have also adopted RWSE over RWPE. In our work, we also use RWSE, the widely accepted term.

In conclusion, both RWSE and RWPE are widely recognized and used interchangeably in the academic community to refer to the same PE method (diagonal of the $k$-steps random-walk matrix).

## H   Limitations

We employ Metis in conjunction with the N2C-Attn module for Cluster-GT. However, Metis is a non-learnable graph partitioning algorithm that provides hard assignments for node clustering.

This poses a limitation as it restricts the flexibility of node groupings, potentially impacting the adaptability of the model in dynamic or complex network scenarios. For future enhancements, exploring combinations of N2C-Attn with other learnable graph partitioning algorithms capable of generating soft assignments may be an interesting direction. Additionally, aspects such as robustness and explainability also warrant further investigation to ensure reliability in real-world settings.

## I Potential Impacts

The proposed Node-to-Cluster Attention mechanism introduces a novel approach to information exchange between node clusters. Our research underscores the importance of incorporating diverse strategies for interactions at both the node and cluster levels. This perspective can be integrated with many existing node clustering-based graph learning methods, enhancing their efficacy and adaptability. Moreover, our experimental validations reveal that the method of interaction between clusters significantly impacts model performance. While current research primarily focuses on how to partition clusters within graphs, our findings suggest a promising direction for future work: optimizing the methods of interaction between clusters. Such advancements could unlock new possibilities for cluster-level graph learning, potentially leading to more robust and sophisticated models that better capture the complexities of large-scale networks.

