# OpenReview forum: "Cluster-wise Graph Transformer with Dual-granularity Kernelized Attention"
_NeurIPS.cc/2024/Conference — NeurIPS 2024 spotlight_

### Official Review · Reviewer_7EZb · 2024-07-04

**Soundness:** 3
**Presentation:** 4
**Contribution:** 2
**Rating:** 6
**Confidence:** 4

**Summary:**

This paper proposed a novel attention method for graphs, focusing on different resolutions of nodes. Instead of attention calculation on the coarsened graph, it views the clustered nodes as sets, and calculates the attention between cluster and node level. Furthermore, it leverages the kernel method for more efficient attention calculation.

**Strengths:**

- Overall, the writing is good and clear.
- The methodology is clear, and the illustration is great.
- The experiment results seem strong.

**Weaknesses:**

- The only novelty is the dual granularity/resolution/hierarchy attention. The kernelization and the multi-kernel are already investigated.
- The graph datasets in the experiments are small. Since the attention can be kernelized and is more efficient, there is no reason not to aim for larger graphs.

**Questions:**

- How do you pick the number of clusters for metis algorithm?
- What properties does the assignment matrix C hold? For example, do rows sum up to 1?
- Intuitively I don't understand why the number of node queries is the same as the number of clusters, not number of nodes. In the illustration, why 3 qs not 9?

**Limitations:**

As mentioned by the authors, the method relies on metis algorithm, which is not flexible enough. Ideally it should be compatible with any graph partitioning algorithms.

---

> ### Author Rebuttal · Authors · 2024-08-06
>
> Thank you for the valuable questions. We provide the following detailed responses to your major concerns.
>
> > **Q1. The only novelty is the dual granularity/resolution/hierarchy attention. The kernelization and the multi-kernel are already investigated.**
>
> We acknowledge the reviewer's comment regarding the familiarity of kernelization and multi-kernel methods. However, we would like to further elucidate the novelty of our approach:
> 1. **Motivation**: In the context where most node clustering-based methods utilize a graph coarsening pipeline—which has its limitations (as illustrated in Fig. 1)—we propose a novel approach for propagating messages between clusters without compressing them. This method allows for deeper interactions between cluster-level and node-level information (as shown in Sec. 3.3).
> 2. **Application of Multi-Kernel Learning (MKL)**: While multi-kernel learning is a mature mathematical tool, we are the first to apply the MKL approach to integrate node-level and cluster-level information.
> 3. **Implementation**: While kernelization is commonly used to accelerate attention computations, our approach integrates kernelization with a message-passing framework. accelerating the attention computation by propagating the keys and values among the clusters (as illustrated in Fig. 2).
>
> In summary, while each mathematical tool has been extensively studied before, our application of these mathematical tools is novel and addresses a previously suboptimal aspect of the graph pooling process.
>
> > **Q2. The graph datasets in the experiments are small. Since the attention can be kernelized and is more efficient, there is no reason not to aim for larger graphs.**
>
> In this paper, our focus is on optimizing clustering-based graph pooling for graph-level tasks. Unlike node-level tasks, graph-level tasks naturally involve graphs of relatively smaller sizes. However, in terms of the number of graphs, the datasets we selected are not "small-scale." (For instance, the OGB MolHIV dataset we used contains 41,127 graphs, which is a considerable size.)
>
> We chose these datasets because they align with the datasets used in the two most relevant baseline works [1,2] that we compare against, ensuring a fair and consistent evaluation of our method.
>
> > **Q3. How do you pick the number of clusters for metis algorithm?**
>
> In our experiments, we set the number of clusters for metis as a hyperparameter from $\\{4, 8, 16, 32\\}$. While this approach of pre-selecting the number of clusters may seem inflexible (as noted in the limitation section), many works in graph pooling (e.g. [1,2,3]) also preset the cluster numbers.
>
> Meanwhile, there are methodologies capable of dynamically adjusting the number of clusters through learning (e.g. [8]). As mentioned in the limitation section, integrating these methods with our N2C-Attn model represents a promising direction for future work.
>
> > **Q4. What properties does the assignment matrix C hold? For example, do rows sum up to 1?**
>
> We clarify that the concept of cluster assignment matrix $\boldsymbol{C}$ is not originally defined in this paper but is instead adopted from reference [4]. It is common for the rows of $\boldsymbol{C}$ to sum up to 1, which can be ensured through a row-wise softmax operation, e.g. [3, 5, 6]. However, configuration where the rows of $\boldsymbol{C}$ do not sum to 1 is also possible, e.g. [7].
>
> Regarding the properties of $\boldsymbol{C}$, there are two main points: Firstly, its shape corresponds to [#Num of nodes, #Num of clusters]. Secondly, the element $\boldsymbol{C}_{ij}$ represents the weight of node $i$ in cluster $j$. These characteristics allow the computation involving $C$, the feature matrix $\boldsymbol{X}$, and the adjacency matrix $\boldsymbol{A}$ to simulate the process of graph coarsening (i.e.  $\boldsymbol{X}^{P} = \boldsymbol{C}^T \boldsymbol{X} ;\quad \boldsymbol{A}^{P} = \boldsymbol{C}^T \boldsymbol{A} \boldsymbol{C}$).
>
> > **Q5. Intuitively I don't understand why the number of node queries is the same as the number of clusters, not number of nodes. In the illustration, why 3 qs not 9?**
>
> We understand the reviewer's concern and offer further clarification here. In the N2C-Attn framework, each node is responsible for providing a pair of keys, and similarly, each cluster supplies a pair of queries (line 121, 128). Thus, a natural setup is to have **the number of keys align with the number of nodes** and **the number of queries align with the number of clusters**.
>
> However, N2C-Attn considers both cluster-level and node-level information. For nodes within the same cluster, their cluster-level information should be identical (line 119). Therefore, the number of cluster-level keys should be equal to the number of clusters.  Thus we have:
> | Type | # Num |
> |-|-|
> | node-level key| Number of nodes
> | cluster-level key| Number of clusters
> | node-level query | Number of clusters
> | cluster-level query | Number of clusters
>
> It's important to note that **node-level** or **cluster-level** specifically indicate the belonging to different feature spaces, $\mathcal{X_N}$ and $\mathcal{X_C}$, respectively. **Node-level queries** are provided by each **cluster**, not by individual nodes. We hope this clears up any confusion for the reviewer.
>
>
> [1]. He et al. A Generalization of ViT/MLP-Mixer to Graphs. ICML 2023.
>
> [2]. Wu et al. Structural entropy guided graph hierarchical pooling. ICML 2022.
>
> [3]. Bianchi et al. Spectral clustering with graph neural
> networks for graph pooling. ICML 2020.
>
> [4]. Liu et al. Graph Pooling for Graph Neural Networks: Progress, Challenges, and Opportunities. IJCAI 2023.
>
> [5]. Ying et al. Hierarchical Graph Representation Learning with Differentiable Pooling. NeurIPS 2018.
>
> [6]. Khasahmadi et al. Memory-based graph networks. ICLR 2020.
>
> [7]. Bacciu et al. A Non-Negative Factorization approach to node pooling in Graph Convolutional Neural Networks. AIIA 2019.
>
> [8]. Song et al. Graph Parsing Networks. ICLR 2024.

---

> > ### Comment · Reviewer_7EZb · 2024-08-11
> >
> > Thank you for your effort in the rebuttal. After reading your response to the reviewers, I would like to raise my score to 6.

---

> > > ### Author Response · Authors · 2024-08-11
> > > **Many thanks**
> > >
> > > Many thanks for the positive feedback. We are dedicated to continual improvement. We also wish to thank the reviewer for improving the quality of our paper.

---

### Official Review · Reviewer_2taa · 2024-07-10

**Soundness:** 3
**Presentation:** 3
**Contribution:** 3
**Rating:** 7
**Confidence:** 3

**Summary:**

This paper considers graph transformers. In previous methods that consider cluster info, node clusters are pooled, which may lose node level information. This paper proposes node-to-cluster attention, where the nodes in the clusters are not compressed, and each cluster can interact with every node in other clusters. It proposed efficient formulation of node-to-cluster attention, and incorporated into the attention mechanism of a graph transformer. A comparison of the proposed method with existing benchmarks, both GCN and transformers, were conducted over 8 datasets, and the proposed method is shown to outperform.

**Strengths:**

- Even though it is a complicated setup, the paper is well written and illustrated, and somewhat managed to get the setup across.

- Experiment is comprehensive, with both GCN and graph transformer benchmarks, study of the necessity of combining cluster and node level info, and efficiency study. The performance seems good

**Weaknesses:**

- Even though it may be unavoidable, the amount of notations make reading the paper somewhat difficult

- Instead of using metis which is a partitioning algorithm, why not use a graph clustering algorithm? Is it because you like each partition to have an equal number of nodes?

**Questions:**

- In addition to the node-to-cluster attention, should there also be cluster-to-node attention and cluster to cluster attention?

- Figure 3: typo? “positinal”->”positional”

**Limitations:**

discussion in Appendix H.

---

> ### Author Rebuttal · Authors · 2024-08-06
>
> We are grateful for the reviewer's comments and suggestions. Below, we provide detailed responses to address the main concerns.
>
> > **Q1. Even though it may be unavoidable, the amount of notations make reading the paper somewhat difficult.**
>
> We fully understand the reviewer's concern about the number of notations, and we acknowledge that it might make the reading experience challenging. But given the complexity of the algorithm setup, these notations are essential for conveying the technical details accurately.
>
> The idea of our approach, however, is straightforward: we integrate multiple kernel learning within the kernelized attention framework, which facilitates effective information transfer at both cluster and node levels among clusters without resorting to the graph coarsening pipeline.
>
> To enhance readability, we have included visual representations of the methodology in Figures 1, 2, and 3 in the paper. Additionally, we provide here a list of notation and the corresponding description.
>
> | Notation | Description |
> |-|-|
> | $(\mathcal{N}, \mathcal{E}, \mathbf{X}, \mathbf{A})$ | Multi-tuple representing the graph: nodes ($\mathcal{N}$), edges ($\mathcal{E}$), node features ($\mathbf{X}$), adjacency matrix ($\mathbf{A}$). |
> | $(\mathcal{N}^P, \mathcal{E}^P, \mathbf{X}^P, \mathbf{A}^P)$ | Cluster-level (coarsened) graph components.
> | $\mathbf{C}$ | Cluster Assignment Matrix, used to map nodes to clusters, obtained from graph partitioning methods.  |
> | $\mathcal{X}_C, \mathcal{X}_N$| Feature space for cluster-level and node-level attributes. |
> |$k_t$| Node-level key, derived from the $t$-th node's embedding: $k\_t = \mathbf{W}\_k h\_t$, where $h_t$ is the feature of the $t$-th node.   |
> | $K_j$| Cluster-level key, representing collective features of the $j$-th cluster: $K\_j = \mathbf{W}'\_k (\sum\_{s}\mathbf{C}\_{sj}h\_{s})$. |
> |$q_i$| Node-level query for the $i$-th cluster interacting with the node-level key: $q\_i = \mathbf{W}\_q (\sum\_{s} \mathbf{C}\_{si} h\_{s})$. |
> |$Q_i$| Cluster-level query for the $i$-th cluster interacting with the cluster-level key: $Q\_i = \mathbf{W}'\_q (\sum\_{s} \mathbf{C}\_{si} h\_{s})$. |
> |$\kappa_B$| Bi-level kernel used in both N2C-Attn-T and N2C-Attn-L. Combines node and cluster level kernels.  |
> |$\kappa_C$| Cluster-level kernel function comparing cluster-level queries and keys. |
> |$\kappa_N$| Node-level kernel function comparing node-level queries and keys. |
> |$\alpha, \beta$| Learnable parameters in N2C-Attn-L, weighting the influence of cluster-level and node-level kernels, respectively. |
> |$\Phi_\mathrm{B}$| Feature map corresponding to the bi-level kernel $\kappa_B$|
> |$\phi, \psi$| Feature maps corresponding to kernel functions for cluster-level and node-level, respectively.  |
> |$v_t$| Node-level values used in the attention computation. |
> |$\langle \cdot, \cdot \rangle$| Inner product|
> |$\otimes, \oplus$| Operators for the outer product and concatenation|
>
> > **Q2. Instead of using metis which is a partitioning algorithm, why not use a graph clustering algorithm? Is it because you like each partition to have an equal number of nodes?**
>
> Our choice of Metis over other graph clustering algorithms was not due to a specific preference (e.g. each partition to have an equal number of nodes); indeed, any method that generates a cluster assignment matrix could substitute Metis. We acknowledge this interchangeability in line 229 of our paper and in the limitation section.
>
> The main reasons for using Metis in our work are:
>
> 1. We selected Graph-ViT [1] as a primary baseline, which employs Metis for subgraph partitioning. To ensure a fair comparison, we followed this setup.
> 2. Metis is a well-established graph partitioning algorithm [2], recognized for its efficient implementations and applications in graph learning (e.g. [3]).
> 3. Unlike other works that focus on optimizing partitions, our study concentrates on the post-partition phase. Therefore, we preferred a straightforward and common partitioning approach to highlight our advancements in post-partition optimization.
>
> > **Q3. In addition to the node-to-cluster attention, should there also be cluster-to-node attention and cluster to cluster attention?**
>
> This is an intriguing question. We address it in two parts:
>
> - **Cluster-to-cluster attention** already exists. In our paper, we refer to techniques that consider only the aggregate information of clusters as cluster-to-cluster attention, such as GraphViT. A more detailed explanation is provided in Appendix B.
>
> - Regarding **cluster-to-node attention**, this would involve each node aggregating information at the cluster level during its representation update. The viability of this approach depends on whether the information from the cluster is useful for node-level tasks. Given that node-level tasks often require finer-grained information than what clusters provide, this assumption may not always hold, warranting further experiments and analysis.
>
>
> > **Q4. Figure 3: typo? “positinal”->”positional”**
>
> Thank you for pointing out this typo. We will make the necessary corrections.
>
> [1]. He et al. A Generalization of ViT/MLP-Mixer to Graphs. ICML 2023.
>
> [2]. Karypis et al. A fast and high quality multilevel scheme for partitioning irregular graphs. SIAM 1998.
>
> [3]. Chiang et al. Cluster-GCN: An Efficient Algorithm for Training Deep and Large Graph Convolutional Networks. KDD 2019.

---

> > ### Comment · Reviewer_2taa · 2024-08-07
> >
> > Acknowledge the response and thanks the careful reply.

---

> > > ### Author Response · Authors · 2024-08-08
> > > **Many thanks**
> > >
> > > Thank you for your feedback and assistance in improving the quality of our paper. We are dedicated to refining our work further and implementing necessary improvements.

---

### Official Review · Reviewer_Tre7 · 2024-07-10

**Soundness:** 4
**Presentation:** 4
**Contribution:** 3
**Rating:** 7
**Confidence:** 4

**Summary:**

The paper introduces the Node-to-Cluster Attention (N2C-Attn) mechanism, which captures both node and cluster-level information using multiple kernels. N2C-Attn can be implemented in the form of linear-time complexity by a cluster-wise message-passing framework. Based on N2C-Attn, the authors propose a Cluster-wise Graph Transformer (Cluster-GT), and demonstrate it outperforms baselines in graph-level benchmarks.

**Strengths:**

This paper is easy to read, with a well-motivated and appropriately designed method to address the identified problem. Using kernelized attention effectively resolves the complexity issue associated with bi-level attention. The theoretical claim aligns well with the model's motivation and behavior. Extensive and diverse experiments demonstrate the model's superior performance and efficiency.

**Weaknesses:**

- The paper emphasizes "without resorting to the graph coarsening pipeline," suggesting that the proposed method eliminates graph coarsening in their model pipeline. However, it actually employs a graph coarsening method (i.e., METIS). While it is true that the proposed method maintains clusters uncompressed, unlike traditional methods that typically coarsen each cluster into a single embedding, this statement might be misleading to readers. The authors should revise this expression to prevent any potential misunderstanding and avoid overselling the method.
- I suggest authors discuss the existing research on GNNs with graph coarsening to capture broader structure information (e.g., higher-order structures or long-range dependencies). Representative models are listed below:
  1. Fey, M., Yuen, J. G., & Weichert, F. (2020). Hierarchical inter-message passing for learning on molecular graphs. arXiv preprint arXiv:2006.12179.
  1. Zhang, Z., Liu, Q., Hu, Q., & Lee, C. K. (2022). Hierarchical graph transformer with adaptive node sampling. Advances in Neural Information Processing Systems, 35, 21171-21183.
  1. Liu, C., Zhan, Y., Ma, X., Ding, L., Tao, D., Wu, J., & Hu, W. (2023). Gapformer: Graph Transformer with Graph Pooling for Node Classification. In IJCAI (pp. 2196-2205).
  1. Fu, D., Hua, Z., Xie, Y., Fang, J., Zhang, S., Sancak, K., ... & Long, B. (2024) VCR-Graphormer: A Mini-batch Graph Transformer via Virtual Connections. In The Twelfth International Conference on Learning Representations.
  1. Kim, D., & Oh, A. (2024) Translating Subgraphs to Nodes Makes Simple GNNs Strong and Efficient for Subgraph Representation Learning. In Forty-first International Conference on Machine Learning.

**Questions:**

- Can the authors rename RWSE to RWPE, following the original work?

**Limitations:**

The authors adequately addressed the limitations.

---

> ### Author Rebuttal · Authors · 2024-08-06
>
> We appreciate the reviewer's comments and suggestions. Here are our detailed responses to your main concerns.
>
> > **Q1. The paper emphasizes "without resorting to the graph coarsening pipeline," suggesting that the proposed method eliminates graph coarsening in their model pipeline. However, it actually employs a graph coarsening method (i.e., METIS). While it is true that the proposed method maintains clusters uncompressed, unlike traditional methods that typically coarsen each cluster into a single embedding, this statement might be misleading to readers. The authors should revise this expression to prevent any potential misunderstanding and avoid overselling the method.**
>
> We thank the reviewer for pointing out this issue. Indeed, the METIS algorithm includes the coarsening (and uncoarsening) process. Thus, strictly speaking, there is a coarsening operation during the partition phase in our implemented cluster-GT.
>
> The point we wish to emphasize is that our work primarily focuses on the post-partition phase (line 227), while the methods used in the partition phase are not our central focus. We can replace Metis with other methods that do not involve coarsening. The proposed approach itself (i.e. N2C-Attn) achieves the objective of "without resorting to the graph coarsening pipeline".
>
> We thank for the reviewer's suggestion. We will revise our manuscript to clarify this point and avoid any potential overstatement.
>
>
> > **Q2. I suggest authors discuss the existing research on GNNs with graph coarsening to capture broader structure information (e.g., higher-order structures or long-range dependencies). Representative models are listed below:**
> >
> >[5]. Fey et al. Hierarchical inter-message passing for learning on molecular graphs. arXiv:2006.12179.
> >
> >[6]. Zhang et al. Hierarchical graph transformer with adaptive node sampling. NeurIPS 2022.
> >
> >[7]. Liu et al. Gapformer: Graph Transformer with Graph Pooling for Node Classification. IJCAI 2023.
> >
> >[8]. Fu et al. VCR-Graphormer: A Mini-batch Graph Transformer via Virtual Connections. ICLR 2024.
> >
> >[9]. Kim et al. Translating Subgraphs to Nodes Makes Simple GNNs Strong and Efficient for Subgraph Representation Learning. ICML 2024.
>
> Thank you for your suggestion and the provided references. Here, we briefly discuss the literature you mentioned and will include a more comprehensive discussion on related work in our paper.
>
> [5] utilizes a dual-graph structure, employing a hierarchical message passing strategy between a molecular graph and its junction tree to facilitate a bidirectional flow of information. This concept of interaction between the coarsened graph (clusters) and the original graph (nodes) is similar to our N2C-Attn. However, the difference lies in [5]'s approach to propagating messages between clusters and nodes, whereas N2C-Attn integrates cluster and node information directly in the attention calculation using a multiple-kernel method.
>
> [6] introduces a novel node sampling strategy as an adversarial bandit problem and implements a hierarchical attention mechanism with graph coarsening to address long-range dependencies efficiently. [7] uses graph pooling to coarsen nodes into fewer representatives, focusing attention on these pooled nodes to manage scalability and computational efficiency. Nonetheless, [6,7] still follow a graph coarsening pipeline, i.e., computing attention on the pooled graph.
>
> [8] focuses on challenges in mini-batch training, proposing to rewire graphs by introducing multiple types of virtual connections through structure- and content-based super nodes. This approach differs from our study, which deals with information propagation between different clusters in the whole graph. [9] introduces the SubgraphTo-Node (S2N) translation method, coarsening subgraphs into nodes to improve subgraph representation learning. While innovative for subgraph classification, it follows the graph coarsening pipeline and does not align directly with the broader graph-level tasks targeted in our research.
>
> > **Q3. Can the authors rename RWSE to RWPE, following the original work?**
>
> We appreciate the reviewer's attention to this interesting detail. Here, we provide a brief investigation into the naming of RWPE.
>
> In the original paper that introduced RWPE [1], the term "Random Walk Positional Encoding (RWPE)" was proposed. This paper utilized the self-landing probability of nodes in a random walk to capture neighborhood structural information.
>
> Subsequently, an influential work in the graph transformer domain [2] made a clear distinction between two types of encodings for structure and position, naming them Positional Encoding (PE) and Structural Encoding (SE). Positional encodings are intended to provide an understanding of a node's position within the graph, while Structural encodings aim to embed the structure of graphs or subgraphs, enhancing the expressivity and generalizability of GNNs.
>
> Interestingly, [2] argues that the Random Walk Positional Encoding (RWPE) proposed in [1] actually serves as a Structural Encoding (SE). Based on our investigation, it is likely that **[2] began using the term RWSE instead of RWPE**. Many subsequent studies, (likely influenced by [2],) such as [3, 4], have also adopted RWSE over RWPE. In our work, we also use RWSE, the widely accepted term.
>
> In conclusion, both RWSE and RWPE are widely recognized and used interchangeably in the academic community to refer to the same encoding method (Diagonal of the $m$-steps random-walk matrix). We will include this brief investigation on the evolution of the term RWPE(RWSE) in our paper.
>
> [1]. Dwivedi et al. Graph Neural Networks with Learnable Structural and Positional Representations. ICLR 2022.
>
> [2]. Rampášek et al. Recipe for a General, Powerful, Scalable Graph Transformer. NeurIPS 2023.
>
> [3]. Shirzad et al. Exphormer: Sparse Transformers for Graphs. ICML 2023.
>
> [4]. He et al. A Generalization of ViT/MLP-Mixer to Graphs. ICML 2023.

---

> ### Comment · Reviewer_Tre7 · 2024-08-09
>
> Thank you for your detailed comments. I will support this paper's acceptance (changed score from 6 to 7).

---

> > ### Author Response · Authors · 2024-08-09
> > **Many thanks**
> >
> > Many thanks for your positive feedback. We are committed to further refining our work and making the necessary improvements to address any concerns. Thank you for the opportunity to enhance the quality of our research.

---

### Official Review · Reviewer_azHs · 2024-07-26

**Soundness:** 3
**Presentation:** 3
**Contribution:** 3
**Rating:** 7
**Confidence:** 5

**Summary:**

The paper proposes an attention-based methodology for supervised graph classification and regression. It adopts a pipeline similar to GraphViT, involving graph partition, cluster-wise representation learning, and aggregation. However, the core mechanism for learning cluster-wise representations is novel. Specifically, it introduces an inter-cluster attention and a cluster-to-node attention, using both attention maps to formulate the attention weights that aggregate node features into the query cluster's representation. This method is evaluated on eight graph classification/regression benchmarks, demonstrating promising results compared to the listed baselines.

**Strengths:**

1. The paper presents a clear and well-supported motivation for the proposed methodology, with technical details that are thoroughly demonstrated.

2. The methodology is novel, and considering both the features of each node and the cluster it is affiliated with when calculating the attention weight is highly reasonable.

3. Experimental results demonstrate the effectiveness of the method. Additionally, the exploratory study on combination weights validates the significance of introducing the learnable bias of the affiliated cluster into the attention weights.

**Weaknesses:**

1. The comparison with graph pooling methods misses MVPooL [1], a recent baseline that has achieved higher accuracy on these benchmarks.

2. Typo: the expression of the attention score between the $i$-th cluster and the $t$-th node in the $j$-th cluster should not contain the value $v_t$.

3. Suggestion: in Figure 2, combining Step 1 with Step 4 might provide a clearer illustration. Only with the presence of queries, keys, and values can the result of an attention operation be in the form of an aggregated representation; the result involving only keys and values is not well-defined. Additionally, combining Step 1 with Step 4 would better match the computation order of Equation 11.

**Reference**

[1] Zhang, Zhen, et al. "Hierarchical multi-view graph pooling with structure learning." IEEE Transactions on Knowledge and Data Engineering 35.1 (2021): 545-559.

**Questions:**

1. Could the authors provide a heuristic explanation for why the cluster-level attention outperforms node-level attention on many benchmarks (as shown in Figure 5)? Cluster-level attention assigns the same weight to all nodes within a cluster, while node-level attention allows more flexible integration between cluster and node representations. It is not immediately clear why a more rigid approach would outperform a more flexible one.

---

> ### Author Rebuttal · Authors · 2024-08-06
>
> We sincerely appreciate the reviewer's feedback. We provide details to clarify the reviewer's major concerns.
>
> > **Q1. The comparison with graph pooling methods misses MVPooL [1], a recent baseline that has achieved higher accuracy on these benchmarks.**
>
> We appreciate the reminder from the reviewer. MVPooL [1] is a compelling method in the domain of graph pooling. Although the datasets involved in the experiments of [1] partially overlap with those used in our work, we did not reference the results from MVPooL in our paper due to differences in experimental settings.
>
> We conduct a comparison of MVPooL under the same experimental setup as described in Section 5.1. (due to time limits, we did not conduct a hyperparameter search and instead used the default hyperparameters provided in the code from [1]). ClusterGT achieved relatively superior results on 5 out of the 6 datasets.
>
> | Model|IMDB-BINARY|IMDB-MULTI|COLLAB|MUTAG|PROTEINS|D&D|
> |-|-|-|-|-|-|-|
> | MVPool     | 72.87±0.69    | 51.04±0.79    | 80.88±0.34    | 82.73±1.21    | 75.15±0.70    | 77.32±0.49    |
> | Cluster-GT | 75.10±0.84    | 52.13±0.78    | 80.43±0.52    | 87.11±1.37    | 76.48±0.86    | 79.15±0.63    |
>
>
>
>
> > **Q2. Typo: the expression of the attention score between the $i$-th cluster and the $t$-th node in the $j$-th cluster should not contain the value.**
>
> Thank you for pointing out this typo. Indeed, the attention score between the $i$-th cluster and the $t$-th node in the $j$-th cluster should be:$
> \frac{\mathbf{A}\_{i, j}^P \mathbf{C}\_{tj} \kappa\_{\mathrm{B}}(\\{Q\_i, q\_i\\},\\{K\_j, k\_t\\})}{\sum\_j \mathbf{A}\_{i, j}^P \sum\_t \mathbf{C}\_{tj} \kappa\_{\mathrm{B}}(\\{Q\_i, q\_i\\},\\{K\_j, k\_t\\})}
> $ . We will make the necessary corrections.
>
> > **Q3.  Suggestion: in Figure 2, combining Step 1 with Step 4 might provide a clearer illustration. Only with the presence of queries, keys, and values can the result of an attention operation be in the form of an aggregated representation; the result involving only keys and values is not well-defined. Additionally, combining Step 1 with Step 4 would better match the computation order of Equation 11.**
>
> We thank for the reviewer's suggestion. However, there might be a slight misunderstanding here. In fact, the result involving only keys and values **is well-defined** (i.e $\sum_t\psi(k_t) v_t$ in the paper, where $\psi(k_t)\in \mathbb{R}^{d_k\times1}$ and $v_t\in\mathbb{R}^{1\times d_v}$). The kernelized softmax trick aggregates keys and values first, and then computes them with queries to reduce computational complexity (e.g. [2,3,4]).
>
> Therefore, Step 1 and Step 4 can be separated. During our implementation, we follow the sequence of Step 1, 2, 3, and 4 as illustrated in Fig. 2, which aligns with the computational process introduced in Section 3.2 and Equation 14.
>
>
> > **Q4.  Could the authors provide a heuristic explanation for why the cluster-level attention outperforms node-level attention on many benchmarks (as shown in Figure 5)? Cluster-level attention assigns the same weight to all nodes within a cluster, while node-level attention allows more flexible integration between cluster and node representations. It is not immediately clear why a more rigid approach would outperform a more flexible one.**
>
> We appreciate the reviewer for highlighting this interesting point.
>
> Firstly, we would like to clarify that: cluster-level attention does not always outperform node-level attention. In fact, as shown in Figure 5, among the four datasets analyzed, cluster-level attention performs better only on IMDB-Binary and IMDB-Multi, whereas node-level attention excels on PROTEINS and D&D. In summary, both cluster-level and node-level attentions have their merits: cluster-level attention is more suitable for social network datasets, while node-level attention fits better with bioinformatics datasets. This may be attributed to the structured nature in social network datasets like IMDB where cluster-level patterns are more pronounced.
>
> Moreover, although node-level attention is indeed more flexible than cluster-level attention, this does not necessarily mean that it outperforms the latter. Cluster-level attention effectively utilizes the auxiliary information of node cluster assignment. Despite being coarser in granularity compared to node-level information, this cluster-level information can be helpful and less noisy, particularly for the graph-level tasks we study in this paper.
>
> Lastly, this issue can also be analyzed from the perspective of the impact of introducing additional constraints on model regularization: while cluster-level attention is not as flexible as node-level attention, the additional constraint within cluster-level attention might help prevent overfitting, enhancing its robustness and generalization capabilities.
>
>
> [1] Zhang et al. Hierarchical multi-view graph pooling with structure learning. IEEE TKDE 2021.
>
> [2]. Katharopoulos et al. Transformers are rnns: Fast autoregressive transformers with linear attention. ICML 2020.
>
> [3]. Wu et al. NodeFormer: A Scalable Graph Structure Learning Transformer for Node Classification. NeurIPS 2022.
>
> [4]. Huang et al. Tailoring Self-Attention for Graph via Rooted Subtrees. NeurIPS 2023.

---

> > ### Comment · Reviewer_azHs · 2024-08-07
> >
> > Thank you for the detailed reply. My concerns are addressed and I will remain my support for the acceptance of the paper, with the score adjusted upward for one point. Best of luck.

---

> > > ### Author Response · Authors · 2024-08-08
> > > **Many thanks**
> > >
> > > Many thanks for the positive feedback. We remain dedicated to ongoing improvement. And we thank the reviewer for helping us improve the quality of our paper.

---

### Author Rebuttal · Authors · 2024-08-06

We extend our sincere gratitude to the reviewers for their insightful feedback.

We are delighted to see the comments regarding the paper's **presentation**: "The paper is well written and illustrated" (Reviewer 2taa), its **motivation**: "The paper presents a clear and well-supported motivation for the proposed methodology" (Reviewer azHs), its **theoretical analysis**: "The theoretical claim aligns well with the model's motivation and behavior." (Reviewer Tre7), and its **experimental setup**: "Extensive and diverse experiments demonstrate the model's superior performance and efficiency." (Reviewer Tre7).

We would like to reiterate the main contributions of our work here:
1. **Motivation:** Current clustering-based graph pooling methods primarily follow a graph coarsening pipeline, which has its limitations (as illustrated in Fig. 1). We propose a method that facilitates information propagation between clusters without compressing them. This approach allows for a deep interaction between cluster-level and node-level information (as shown in Sec. 3.3).
2. **Technical Contribution:** We enhance the kernelized attention framework by integrating Multi-Kernel Learning (MKL), allowing for a more nuanced merging of information at both cluster-level and node-level granularities. Additionally, leveraging kernelization techniques, we implement an efficient attention computation method by propagating the aggregated keys and values among clusters.

In the following individual responses, we have addressed the main concerns raised by the reviewers. We are grateful for the reviewers' detailed suggestions. If there are any further questions or comments you wish to discuss, please do not hesitate to reach out.

---

### Decision · Program_Chairs · 2024-09-25

**Decision:**

Accept (spotlight)

**Comment:**

The paper proposes a new attention mechanism which captures information at both node and cluster levels. The proposed mechanism is novel and it performs well in practice.